# Demographic, Epidemiologic, and Clinical Characteristics of Human Monkeypox Disease Pre- and Post-2022 Outbreaks: A Systematic Review and Meta-Analysis

**DOI:** 10.3390/biomedicines11030957

**Published:** 2023-03-20

**Authors:** Hossein Hatami, Parnian Jamshidi, Mahta Arbabi, Seyed Amir Ahmad Safavi-Naini, Parisa Farokh, Ghazal Izadi-Jorshari, Benyamin Mohammadzadeh, Mohammad Javad Nasiri, Milad Zandi, Amirhossein Nayebzade, Leonardo A. Sechi

**Affiliations:** 1Department of Public Health, School of Public Health and Environmental and Occupational Hazards Control Research Center, Shahid Beheshti University of Medical Sciences, Tehran 1985717443, Iran; hatami@sbmu.ac.ir; 2Department of Microbiology, School of Medicine, Shahid Beheshti University of Medical Sciences, Tehran 1985717443, Iran; mahtaarbabi@sbmu.ac.ir (M.A.); parisa.farokh@sbmu.ac.ir (P.F.); benjaminmohammadzadeh@gmail.com (B.M.); mj.nasiri@hotmail.com (M.J.N.); 3Research Institute for Gastroenterology and Liver Diseases, Shahid Beheshti University of Medical Sciences, Tehran 1985717443, Iran; sdamirsa@ymail.com; 4Student Research Committee, School of Medicine, Shahid Beheshti University of Medical Sciences, Tehran 1985717443, Iran; ghazal.izadi.j@gmail.com; 5Department of Virology, School of Public Health, Tehran University of Medical Sciences, Tehran 1417613151, Iran; miladzandi416@gmail.com; 6Urology and Nephrology Research Center, Shahid Beheshti University of Medical Sciences, Tehran 1985717443, Iran; nayebzade.amir@gmail.com; 7Department of Biomedical Sciences, University of Sassari, 07100 Sassari, Italy; 8SC Microbiologia e Virologia, Azienda Ospedaliera Universitaria, 07100 Sassari, Italy

**Keywords:** monkeypox, mpox, outbreak, clinical manifestations, rash, lymphadenopathy, mortality

## Abstract

(1) Background: In early May 2022, an increasing number of human monkeypox (mpox) cases were reported in non-endemic disparate regions of the world, which raised concerns. Here, we provide a systematic review and meta-analysis of mpox-confirmed patients presented in peer-reviewed publications over the 10 years before and during the 2022 outbreak from demographic, epidemiological, and clinical perspectives. (2) Methods: A systematic search was performed for relevant studies published in Pubmed/Medline, Embase, Scopus, and Google Scholar from 1 January 2012 up to 15 February 2023. Pooled frequencies with 95% confidence intervals (CIs) were assessed using the random or fixed effect model due to the estimated heterogeneity of the true effect sizes. (3) Results: Out of 10,163 articles, 67 met the inclusion criteria, and 31 cross-sectional studies were included for meta-analysis. Animal-to-human transmission was dominant in pre-2022 cases (61.64%), but almost all post-2022 reported cases had a history of human contact, especially sexual contact. The pooled frequency of MSM individuals was 93.5% (95% CI 91.0–95.4, I^2^: 86.60%) and was reported only in post-2022 included studies. The male gender was predominant in both pre- and post-2022 outbreaks, and the mean age of confirmed cases was 29.92 years (5.77–41, SD: 9.38). The most common clinical manifestations were rash, fever, lymphadenopathy, and malaise/fatigue. Proctalgia/proctitis (16.6%, 95% CI 10.3–25.6, I^2^: 97.76) and anal/perianal lesions (39.8%, 95% CI 30.4–49.9, I^2^: 98.10) were the unprecedented clinical manifestations during the 2022 outbreak, which were not described before. Genitalia involvement was more common in post-2022 mpox patients (55.6%, 95% CI 51.7–59.4, I^2^: 88.11). (4) Conclusions: There are speculations about the possibility of changes in the pathogenic properties of the virus. It seems that post-2022 mpox cases experience a milder disease with fewer rashes and lower mortality rates. Moreover, the vast majority of post-2022 cases are managed on an outpatient basis. Our study could serve as a basis for ongoing investigations to identify the different aspects of previous mpox outbreaks and compare them with the current ones.

## 1. Introduction

Human monkeypox (mpox) is a zoonotic infectious disease caused by the monkeypox virus of the Poxviridae family [1]. The name monkeypox originated in a Danish laboratory in 1958, and the first human case of mpox was detected two decades later in an infant from the Democratic Republic of Congo (DRC) [2,3]. After the eradication of smallpox in 1980, mpox became the most threatening Poxviridae family member, causing 300–500 million deaths [4]. Since then, most cases have been sporadic and diagnosed in African countries for many years. In 2003, the first confirmed mpox cases outside of Africa were reported in the United States [5]. Mpox was classified as endemic in Nigeria in 2017, and incidence rates increased significantly in recent years [6,7]. Since early May 2022, an increasing number of mpox cases have been reported in non-endemic regions of the world. This concurrent high incidence of mpox cases in very different geographic regions was a concern of the World Health Organization (WHO). Interestingly, most of the reported patients had a history of sexual contact and mainly, but not exclusively, involved men who have sex with men (MSM) [8].

By 26 February 2023, the World Health Organization (WHO) reported 86,127 confirmed cases and 97 deaths in 110 countries where mpox was previously rare [9]. This rapid global spread of infection warranted public health measures, while the COVID-19 pandemic is still ongoing [10]. Currently, the Emergency Committee WHO has called on scientists worldwide to collaborate on mpox to better understand the disease and prevent further harm [11].

In the current situation, the correct and rapid identification of patients is necessary to achieve control measures to prevent the spread of the disease and to improve patient care. Given the possibility that the route of transmission, demographic characteristics, and clinical manifestations of the disease may change in the upcoming epidemics, conducting a systematic review and meta-analysis of previous information can provide a basis for a better comparison of the current outbreak with previous outbreaks. 

Here, we provide a comprehensive systematic review and meta-analysis of mpox-confirmed patients presented in peer-reviewed publications over the 10 years before and during the 2022 outbreak from demographic, epidemiologic, and clinical perspectives to highlight the differences in mpox characteristics over two periods.

## 2. Methods

This review conforms to the “Preferred Reporting Items for Systematic Reviews and Meta-Analyzes” (PRISMA) statement [12].

### 2.1. Search Strategy and Selection Criteria

A systematic search was performed for relevant studies published in Pubmed/Medline, Embase, and Scopus, from 1 January 2012 up to 15 February 2023. We also searched Google Scholar for relevant grey literature.

Articles that contained the following keywords in the title or abstracts were selected: “Monkeypox” OR “Monkeypox virus”. The records found during the database search were merged using EndNote X8 (Thomson Reuters, New York, NY, USA), and duplicates were removed. Two reviewers (MA and PF) independently reviewed the records by title and abstract to exclude those not related to the present study. The full text of potentially eligible data sets was retrieved and assessed separately by two other reviewers (SAASN, GIJ). In each step, contraries were discussed with a third viewer (PJ). Inclusion criteria were as follows: (i) original descriptive studies that (ii) included confirmed mpox cases and (iii) contained sufficient data about the epidemiologic, demographic, and clinical characteristics of the patients. Exclusion criteria were as follows: review articles, duplicate publications, animal studies, in vitro/in vivo studies, case reports, case series with fewer than 3 cases, case-control studies, conference abstracts, news, commentaries, epidemiologic reports, book sections, modeling studies, molecular and genetic studies, and articles for which full text could not be found, and those with no relevant data. Studies with insufficient information on patient characteristics and outcomes were also excluded. Only English-language articles were considered. In the next step, the included cross-sectional studies were considered for further meta-analysis.

### 2.2. Data Extraction 

Data on first author’s name, first author’s country, location of the outbreak or reported cases, cohort/time of the outbreak, time of publication, type of study, viral clade, mean age, sex, nationality, number of confirmed, probable, suspected, primary, and secondary cases according to the study’s case definition, epidemiologic history (travel, animal contact, human contact, occupation, mentioned route of transmission, and mentioned behavioral risk factors), smallpox vaccination status, comorbidities, diagnostic confirmatory criteria, clinical manifestations, rash localization, rash severity, lymphadenopathy localization, complications, management status, outcome, duration of disease, and patients’ incubation period were extracted for further analysis. Selected data were extracted from the full texts of eligible publications by all team members.

### 2.3. Quality Assessment

The critical appraisal checklist for systematic reviews provided by the Joanna Briggs Institute (JBI) was used to assess the quality of the studies. In addition, included cross-sectional studies were reassessed using the critical appraisal checklist for prevalence studies provided by the JBI for inclusion in the meta-analysis [13].

### 2.4. Data Synthesis and Analysis

Statistical analyzes were performed using Comprehensive Meta-Analysis software, version 2.0 (Biostat Inc., Englewood, NJ, USA). Pooled frequencies with 95% confidence intervals (CIs) were assessed using the random or fixed effect model due to the estimated heterogeneity of the true effect sizes. Heterogeneity between studies was assessed based on Cochran’s Q and the I^2^ statistic. Publication bias was statistically assessed using Begg’s tests (*p* < 0.05 was considered to indicate statistically significant publication bias) [14].

## 3. Results

As shown in Figure 1, our initial search yielded 10,163 articles, of which 6243 duplicate articles were excluded, and 202 articles were selected following title and abstract screening. After the full-text screening, 67 articles were included, of which 31 cross sectional studies were considered for further meta-analysis.

### 3.1. Quality of the Included Studies

The JBI checklist for prevalence studies showed that the cross-sectional studies included in the meta-analysis had a low risk of bias (Table 1).

### 3.2. Characteristics of the Included Studies

Almost half of the articles were cross-sectional studies (*n* = 32), and the remaining were in this order: 14 case series, 7 letters, 5 brief reports, 4 rapid communications, 3 short communications, and 2 correspondences (Table 2). Most of the included studies (41/67) were conducted during the 2022 outbreak and were located mostly in Spain (12/41), the USA (8/41), and the UK (5/41). In contrast, most of the pre-2022 outbreak studies were located in Africa (23/26), especially the Central African Republic (CAR), Democratic Republic of Congo (DRC), and Nigeria (Figure 2). The total number of cases was 36,682, of which 33,673 were confirmed cases. Most of our studied population was reported during the 2022 outbreak (89.52%, 30,145/33,673) (Table 2).

Although the case definition of mpox differed among pre-2022 studies, the criteria of studies had roughly similar characteristics. The detection of viral DNA via PCR (RT-PCR) or virus isolation from patient samples was used to confirm mpox cases. Probable and suspected case definitions were more varied. A probable case was defined when confirmatory laboratory tests were not available and the probability of mpox disease was high (epidemiologic risk factors or contact with known cases). Suspected cases were defined when an unexplained, sudden-onset fever was followed by a rash accompanied by lymphadenopathy or involvement of the palms or soles. Table 3 describes the different approaches used by the included studies, compared with the 2022 WHO and CDC case definitions.

According to the similarity of clinical manifestations of mpox and varicella zoster disease, we excluded 286 MPX/VZV coinfected individuals from our study population and our analysis was based on 30,728 confirmed mpox-only cases.

Based on the data available to us, the West African strain was the most common clade of the virus in both pre- and post-2022 mpox outbreaks (Table 2).

### 3.3. Characteristics of the Study Population (Confirmed Cases)

Before the 2022 outbreak, most of the cases (61.64%) were primary, having a history of animal exposure (animal-to-human transmission) and the rest (38.35%) were secondary cases (human-to human-transmission), of which most were among households or healthcare workers or had a travel history to endemic areas. However, during the 2022 outbreak, almost all of the reported cases were deemed secondary cases. Most of the cases had a history of skin-to-skin contact with symptomatic or asymptomatic patients, mainly sexual contact or attendance in mass gathering events (e.g., Pride) or indirect contact with contaminated objects in locations where outbreaks of mpox were ongoing (Table 2 and Table 4). The pooled frequency of MSM individuals in our included cross-sectional studies was 93.5 (95% CI 91.0–95.4, I^2^: 86.60%), and it was reported only in post-2022 studies (Table 6). 

The pooled frequency of the male gender was 98.7% (95% CI 97.1–99.4, I^2^: 92.19%) in post-2022 studies, which is higher than the previous ones (59.1%, 95% CI 54.6–63.6, I^2^: 77.35%) However, there was evidence of significant publication bias (Begg’s test *p* < 0.05) regarding the gender-specific post-2022 data (Table 6). The mean age of the confirmed cases was 29.92 years (5.77–41, SD: 9.38) (Table 4).

Most of the cases in pre-2022 outbreaks were from Africa or had a history of travel to African endemic countries. In contrast, the majority of the post-2022 cases were reported from non-endemic areas, including America, Europe, and Asia (Table 2 and Table 4).

A reverse transcriptase polymerase chain reaction (RT-PCR) test from the skin lesions was the most common diagnostic confirmation tool in our reviewed studies. However, in post-2022 studies, an RT-PCR test through anal, and oropharyngeal swabs was also more common.

Epidemiologic, demographic, and clinical characteristics of the included study populations are defined in Table 4 and Table 5 in detail.

The most common clinical manifestations of mpox during pre- and post-2022 outbreaks were rash, fever, lymphadenopathy, and malaise/fatigue. However, the pooled frequency of each of the above symptoms was lower in post-2022 cases compared to that among the previous ones. More details are available in Table 6.

Proctalgia/proctitis was the unprecedented clinical manifestation of mpox during the 2022 outbreak, which was not described before. We calculated a pooled frequency of 16.6% (95% CI 10.3–25.6, I^2^: 97.76) for proctalgia/proctitis in our post-2022 included studies.

The most frequent rash locations in pre-2022 outbreaks were the head and neck (98.0%, 95% CI 97.1–98.6, I^2^: 45.18%), trunk (95.2%, 95% CI 82.4–98.8, I^2^: 93.32%), upper limbs (94.9%, 95% CI 82.7–98.6, I^2^: 92.92%), and lower limbs (91.0%, 95% CI 71.0–97.7, I^2^: 92.77%). Of note, palmar and/or plantar involvement was reported in 85.9% (95% CI 65.1–95.28, I^2^: 97.77%) and 73.8% (95% CI 55.3–86.5, I^2^: 97.48%) of patients, respectively. Oropharyngeal (52.4%, 95% CI 45.5–59.2, I^2^: 90.31%) and genital (53.5%, 95% CI 36.8–69.5, I^2^: 94.03%) involvement were less common according to our meta-analysis. In contrast, the most common rash locations in post-2022 patients were the genitalia (55.6%, 95% CI 51.7–59.4, I^2^: 88.11), trunk (45.2%, 95% CI 38.6–52.0, I^2^: 96.19), upper limb (41.8%, 95% CI 36.4–47.4, I^2^: 94.37), and anal/perianal area (39.8%, 95% CI 30.4–49.9, I^2^: 98.10). Anal/perianal involvement was not reported in pre-2022 included studies. The pooled frequency of oropharyngeal lesions in post-2022 outbreaks was 18.3% (95% CI 13.2–24.9, I^2^: 95.95%), which was lower than the previous frequency (Table 6). Figure 3 and Figure 4 indicate the pooled frequency of genitalia involvement in pre- and post-2022 included studies. As shown, genitalia involvement was more common during the 2022 outbreak and was reported from much more studies with a greater population (Table 6)

The most commonly involved lymph nodes in patients with LAP were cervical (79.7%, 95% CI 61.6–90.6, I^2^: 97.09) and inguinal lymph nodes (44.1%, 95% CI 30.2–59.1, I^2^: 95.57) in pre- and post-2022 cases, respectively (Table 6).

Several studies reported the disease severity of mpox cases based on the WHO clinical severity score [22] and divided patients into mild (<25 skin lesions), moderate (25–99 skin lesions), severe (100–250 skin lesions), and grave (>250 skin lesions). According to our analysis, most of the pre-2022 patients had moderate skin rash severity (53.6%, 95% CI 46.6–60.5, I^2^: 88.72%). In contrast, most of the post-2022 mpox patients experienced mild skin rash severity (84.9%, 95% CI 71.9–92.5, I^2^: 96.28) (Table 6).

Figure 5 summarizes the clinical features of confirmed mpox cases before and after the 2022 outbreak.

Insufficient data were available on VZV coinfection in mpox-confirmed patients. Four studies investigated the probability of MPOX/VZV coinfection in their patients and only two of them reported the exact coinfected cases [16,22]. Based on these studies, we calculated the frequency of MPOX/VZV coinfection as 21.86% (Table 5 and Table 6).

The pooled frequency of HIV in mpox-confirmed patients was 11.4% (95% CI 0.2–91.3, I^2^: 88.75) and 41.1% (95% CI 35.5–47.0, I^2^: 94.91) among pre- and post-2022 cases, respectively, indicating higher prevalence of HIV in post-2022 mpox patients.

The smallpox vaccination status of mpox patients was considered because of the potential preventive role of the smallpox vaccine in controlling recent outbreaks of mpox infection. Of the pre-2022 confirmed cases with a known smallpox vaccination status, 94.4% (CI 89.4–97.2, I^2^: 86.92%) had no history of smallpox vaccination and 5.6% (CI 2.8–10.6, I^2^: 86.92%) had been previously vaccinated. The pooled frequency of smallpox vaccination in post-2022 cases was much greater than the previous ones (11.0%, 95% CI 8.9–13.6, I^2^: 67.70 vs. 5.6%, 95% CI 2.8–10.6, I^2^: 86.92); however, there was evidence of publication bias in post-2022 studies regarding the vaccination status (Begg’s test *p*-value < 0.05) (Table 6).

Significant complications of the disease were bacterial superinfections and abscesses, keloids, hyperpigmented atrophic scars, hypertrophic scars, alopecia, severe ulcerative proctitis, sepsis, cellulitis, epiglottitis, bronchopneumonia, lymphangitis, balanitis, urinary complications and obstructions, orchiepididymitis, myocarditis, ocular opacities, keratitis, and unilateral or bilateral ocular complications (Table 5).

Only 21 articles reported the incubation period of the disease in their cases (Table 5). However, according to our calculations, the mean incubation period was 7.9 days (1–21, SD: 4.0). Moreover, the disease duration was reported only in 17 studies, and according to our analysis, the mean disease duration was 17.33 days (7–35, SD: 8.40) (Table 5).

The majority of the pre-2022 cases were treated as inpatients (76.2%, 95% CI (30.1–96.0, I^2^: 52.72), while most of the post-2022 cases were managed as outpatients (95.1%, 95% CI 92.9–96.6, I^2^: 87.49) (Table 6).

Regarding patient outcome, our meta-analysis showed a mortality rate of 4.2% (95% CI 1.6–11.0, I^2^: 83.22) in pre-2022 outbreaks and a mortality rate of 0.2% (95% CI 0.1–0.3, I^2^: 19.18) during the 2022 outbreak (Table 6). It is important to note that the outcome of the majority of reported cases before 2022 was unknown. Therefore, it is probable that the pooled mortality rate of pre-2022 cases calculated in our study is higher than the actual rate.

## 4. Discussion

While the COVID-19 pandemic has not yet ended, the world has been confronted with a new health emergency due to the monkeypox virus [86]. Since the eradication of smallpox in 1980, mpox has been the most widespread and, in terms of morbidity and mortality, the most important orthopoxvirus infection in humans [17,20,87,88,89,90,91]. The gradual increase in the number of cases in endemic areas and the increasing reports of mpox outside endemic areas in recent years have raised concern about the epidemic potential of the virus [7,17]. Two major clades of the virus have been identified: the West African clade, which is associated with a mild disease course and lower human-to-human transmission, and the Congo Basin clade, which is associated with severe disease, higher mortality, and greater human-to-human transmission [89,92,93]. According to our study, more than half of the cases reported in the past 10 years were primary cases involving animal exposure. Human-to-human transmission through skin contact or droplet infection in households or among healthcare workers was also included in 38.35% of cases in the studies reviewed. This low frequency of human-to-human transmission can be explained by the dominance of the West African clade in the articles we reviewed.

To date, there is no conclusive evidence of biological or genetic changes in the virus leading to the current resurgence. Moreover, the West African clade has been isolated from some of the cases in the current outbreak [7,94]. However, an analysis by Kugelman et al. indicated gene loss in 17% of the samples from the Democratic Republic of Congo (DRC), which appeared to be associated with an increase in human-to-human transmission [6,95]. There is increasing evidence of a possible new zoonotic reservoir for the virus and human-to-animal transmission outside of endemic areas [96]. In addition, there is the possibility of undetected transmission in the community via an unusual route, possibly sexual transmission, as indicated by the high incidence of the disease in the community of men who have sex with men (MSM) [8,93,96]. Sexual transmission of mpox virus is not a new possibility. Ogoina et al. reported the likelihood of sexual transmission of the virus through close skin-to-skin contact in their young adult patients with genital ulcers in the 2017 Nigerian outbreak [21]. Based on our meta-analysis, the pooled frequency of genital lesions has increased during the 2022 outbreak (55.6% vs. 53.5%). In the current outbreak, the transmission rate among sexually active participants, especially MSM or bisexual individuals, is surprising. For example, Vivancos et al. reported that 83% of confirmed mpox patients in their study were MSM or bisexual [97]. In addition, we found a pooled frequency of 93.5% for MSM individuals in the post-2022 included studies. Therefore, it is very plausible that sexual activity is a possible route of transmission for the disease. An analysis of seminal fluid from some patients has demonstrated the existence of the mpox virus [98]. Bragazzi et al. suggested that the presence of mpox virus in seminal fluid may be due to the systemic spread of the virus, called viremia, as well as defects in the blood–testicular barrier and immune privilege of the testis [98]. However, the question remains about the infectivity of the virus in semen. Further studies are needed to clarify this issue.

The clinical manifestations of mpox are another problem in distinguishing mpox from other similar diseases, such as varicella-zoster virus (VZV) infection, in the current outbreak. According to our analysis, rash and fever were the most common symptoms of mpox, with pooled frequencies of 96.6% and 62.3% in the post-2022 cases, respectively. The timing of fever and rash is one of the most important points in distinguishing varicella zoster from mpox, as in mpox, a high-grade fever usually occurs before the rash, whereas in VZV infection, a mild fever usually occurs simultaneously with the rash [22]. Most patients in the studies we examined had experienced a febrile prodromal stage with or without chills, headache, sore throat, malaise, and myalgia lasting for 1 to 4 days. As with the improvement of prodromal symptoms, the rash progressed slowly, with centrifugal distribution and concentration on the face and extremities. According to our results, before the 2022 outbreak, the rashes occurred most frequently on the head and neck, followed by the trunk, upper limbs, and lower limbs. Our analysis showed that the palms were more frequently affected (85.9%) than the soles, for which a pooled frequency of 73.8% was calculated. However, involvement of the palms (15.4%) and soles (10.6%) was less common in post-2022 studies compared with that in the previous outbreaks. The pooled frequencies we calculated are both lower than those in the study by Osadebe et al., who found involvement of the palms and soles in 91.2% of cases [17]. Involvement of the palms and soles has been noted in other orthopoxvirus infections, such as smallpox. This has also been noted in VZV cases, with a prevalence of 81.3% [17,99]. However, palms and soles are more commonly affected in mpox cases, and Osadebe et al. suggested this sign as one of the 12 signs/symptoms for mpox-specific case investigations [17]. One of the most important differences between mpox and VZV is that in mpox, all lesions are at the same stage and develop slowly in the order of macule, papule, vesicle, pustule, crust, and finally desquamation over 1–2 days [90,100]. In contrast, VZV lesions usually occur in multiple stages on different parts of the body and rapidly develop from a macule to a crust within a day or sooner [22]. The appearance of mpox lesions is another important consideration. According to the studies we reviewed, mpox lesions are firm, deep-seated, monomorphic, and well-circumscribed, with central punctate nodules. In contrast, VZV lesions are superficial and have an irregular border [22].

Lymphadenopathy (LAP) is another common manifestation that can be used as a distinguishing feature for mpox. The pooled frequency of LAP in our study was 80.6% and 55.5% during the pre- and post-2022 outbreak, respectively. LAP is typically not prominent in VZV patients and unlike smallpox, it may occur before or at the onset of rash [7,17,101].

The pooled frequency of genital involvement in our study was 55.6% during the 2022 outbreak which is higher than that in previous reports. Genital ulcer has also been observed in VZV cases, as reported by Osadebe et al., in 14.9% of their patients [17]. However, it occurs more frequently in mpox cases and has a high specificity for mpox [17].

It appears that the disease pattern has changed slightly in the current 2022 outbreak. The CDC states that mpox rashes can begin in the mouth and spread to the face and extremities, including the palms and soles of the feet [93]. Atypical disease courses have also been reported, such as the absence of or minimal prodromal symptoms, isolated lesions confined to the genital/anal area, and initial genital rashes followed by facial and extremity involvement [67,97,98,102,103,104]. Further population-based studies are needed to determine possible changes in the clinical manifestations of mpox disease.

There is conflicting information on the onset of transmissibility in patients. Patients are thought to be infectious from the onset of the rash and during the four-week desquamation phase because of high viral shedding [93]. However, Nolen et al. point out that patients may be infectious even before the onset of the rash [15].

VZV/MPOX coinfection was reported in some of the studies we reviewed. However, the exact mechanism behind this is not yet fully understood. It is not clear whether the existence of two viruses in a host occurs independently or whether one virus influences the pathogenesis of the other [16,105]. Some studies suggest that mpox can directly trigger VZV reactivation [16,22,106]. Evidence for this hypothesis comes from historical case reports of herpes zoster reactivation after smallpox vaccination [107]. Hughes et al. suggested that primary infection with VZV or mpox weakens the patient’s immune system and leads to susceptibility to secondary infections [22]. In endemic areas with a high prevalence of mpox and VZV, the patients are more susceptible to these pathogens as secondary infections. Another hypothesis is that VZV lesions disrupt skin integration, making it easier for mpox virus to invade [22]. We have estimated the prevalence of MPOX/VZV coinfection to be about 22%. However, in reality, it could be higher or lower due to a biased selection of studies or inadequate investigations, respectively. VZV/MPOX coinfected patients are more likely to report symptoms, and their number of lesions is significantly higher compared with those in only VZV patients [22]. On the other hand, the disease is more severe in patients with only mpox virus than in VZV/MPOX-coinfected patients. This may lead to a bias in the selection of reported cases [22]. In the 1980 laboratory protocols of the surveillance program of the WHO, dual testing of VZV and mpox in suspected individuals was not supported unless mpox tests were negative [16,108]. Therefore, it is likely that a larger percentage of mpox-confirmed patients have undiagnosed VZV coinfection and that the calculated incidence is lower than the true rate.

Coinfection of mpox and HIV is another point of challenge that requires further research. The incidence of MPOX/HIV coinfection was 41.1% in our study, according to the available data on post-2022 cases, which is greater than the pre-2022 calculated frequency (11.4%). Most of the pre-2022 cases we studied were reported from the Democratic Republic of Congo and Nigeria, where HIV is endemic. HIV prevalence in Nigeria was reported to be 3.4% in 2018 [21]. Given that HIV testing was not performed in the majority of cases studied, it is highly plausible that the prevalence of MPOX/HIV coinfection is higher than that calculated in our analysis. It appears that HIV-positive mpox patients have more severe disease with higher morbidity and mortality [21]. In the 2017–2018 outbreak in Nigeria, four of seven patients who died were HIV-positive [20]. Ogoina et al. reported that the incidence of genital ulcers, secondary infections, and complications, as well as the greater rash size and longer disease duration, were significantly associated with HIV positivity in mpox cases. In addition, this study showed that the age, sex, hospitalization, and outcome of these patients did not differ significantly from those of HIV-negative mpox-positive patients [56,98]. In the current outbreak, preliminary data suggest that HIV seropositivity and a previous history of sexually transmitted infections may be risk factors for mpox infection [109]. Therefore, we suggest that mpox vaccination should be a priority prevention measure for this population.

The mean age of mpox cases in our study was calculated to be 29.9 years, which is greater than the previous data from the 2010–2019 outbreak investigations, where the median age at disease onset was 21 years [6]. Available demographics from previous outbreaks show a clear shift in the mean age of mpox cases over time. The mean age has increased from 4 years in the 1970s to 10 years in the 2000s and 21 years in the 2010s [6]. Concerning the outbreak in 2022, the current literature gives an average age of 30 years for those affected, which is consistent with our study [98]. This increasing age trend in patients over time appears to be due to decreased immunity to smallpox following the cessation of smallpox vaccination in the 1980s [23,110,111,112]. According to our meta-analysis, only 5.6% and 11% of pre- and post-2022 mpox patients had been previously vaccinated, respectively. In a study by Whitehouse et al., it was suggested that 10% of mpox cases had been vaccinated before mpox infection [23]. There is evidence that historical smallpox vaccination provides cross-protective immunity against other poxviridae family members including cowpox and monkeypox [113,114,115]. The effectiveness of the vaccinia vaccine against mpox is reported in various studies to be 81–85% [90,91,96,113]. However, some studies suggest that smallpox vaccination-induced immunity declines over time, particularly after 20 years of vaccination, but it appears to reduce the morbidity and mortality from mpox disease [18,100,116]. As with the investigation into the 2017 outbreak in the Democratic Republic of the Congo, the mortality rate among unvaccinated mpox-confirmed cases was approximately 9.5% [115]. Vaccination of close contacts has also reduced transmission of the disease in previous outbreaks [93].

The majority of cases reported in our study were male, consistent with previous studies. This may be due to the gender-specific roles in previous outbreaks in endemic areas, as males had greater exposure to animal reservoirs when hunting [117,118,119]. Young men in particular were also affected by the current outbreak [98]. This could be due to their increased sexual activity and participation in festivals such as Pride [98]. However, we found a significant publication bias regarding the gender of the individuals in post-2022 studies. We assume that during the 2022 outbreak, most of the attention has been assigned to screening the MSM community and this has resulted in the mentioned bias and a stigma. Both genders can be infected with mpox. To date, several studies have reported mpox cases in women including old women [120,121]. CDC reported a total of 769 cases of mpox among cisgender women, including 23 (3%) who were pregnant, by 42 public health jurisdictions from 11 May to 7 November 2022 [122]. The clinical features of mpox in females were similar to those described in males, including the presence of anal and genital lesions with marked mucosal involvement. Anatomically, anogenital lesions reflected sexual practices: vulvovaginal lesions predominated in cis women, with anorectal features in trans women [64,123,124,125].

The mortality rate from mpox cases has been reported to be 1–11% in previous outbreaks [7,20,23,101]. However, the mortality rate is highly dependent on medical care, the endemic nature of the disease, the viral clade, and the characteristics of the individuals affected. Bunge et al. analyzed an all-cause mortality rate of 8.7% with a range of 10.6% for the Central African and 3.6% for the West African clade of the virus. This value was 4.6% for the West African clade exclusively in endemic areas [6]. The mortality rate is very low in areas with good medical care, since there were no reported deaths when the disease broke out in the US in 2003 [93,98]. According to patient characteristics, the majority of deaths have been reported in infants, children under 10 years of age, pregnant women, immunocompromised patients (e.g., HIV), unvaccinated individuals, and those who have developed complications [7,87]. As the age at diagnosis increases, an increasing trend in the age at death has also been demonstrated [6]. In our analysis, there was little data on patient treatment status and outcomes in pre-2022 studies. Using the available data, we calculated a pooled mortality rate of 4.2% and 0.2% in pre- and post-2022 mpox cases, respectively. However, we believe that a large amount of missing data and a bias in the selection of reported cases in pre-2022 studies has resulted in the higher calculated mortality rate of previously described cases.

One of the rare but significant mpox complications is ocular involvement. These injuries are divided into (a) more common and benign lesions and (b) less common and vision-threatening sequelae. Conjunctivitis, blepharitis, and photophobia are the most commonly reported uncomplicated manifestations. There are also mpox-related symptoms, such as eye redness, frontal headache, periocular and orbital rash, lacrimation and ocular discharge, subconjunctival nodules, and less frequently, keratitis, corneal ulceration, opacification, and perforation [126]. Ocular manifestations were less common and probably less severe in the current outbreak. The pooled frequencies of ocular involvement, mostly conjunctivitis, were 23.3% and 1.6% in pre- and post-2022 studies, respectively. Observational studies suggest rates of about 1% ocular involvement in the current outbreak, compared with 9–23% in previous outbreaks in endemic countries, which is consistent with our calculated rates [127]. Smallpox vaccination in the past is a protective factor against these complications. Although there is no clear and established treatment, simple therapies, such as regular lubrication and the prophylactic use of topical antibiotics, can be considered for ocular complications of mpox. The timely administration of specific antiviral agents may also be effective in severe cases. Mpox usually has mild-to-moderate severity and a self-limited course. The risk of permanent ocular sequelae and disease morbidity could be reduced if the disease is identified early and appropriately treated [128,129].

## 5. Limitations and Suggestions

One of the major limitations of our study was the insufficient and missing data from the pre-2022 articles reviewed. All of our analyses have been performed based on the available data, and due to the significant amount of unavailable information, especially regarding the pre-2022 included studies, our final statistics may be overstated or understated. Insufficient data on the characteristics of the reported deaths resulted in an inability to stratify the mortality rate by age, sex, immunization status, comorbidities, etc. Further investigation is needed to clarify this issue. We strongly encourage future investigators to provide complete data on the cases that they report. Especially, comprehensive information about dead individuals is urgently needed to reveal high-risk patients. These data are required to prioritize the high-risk individuals for inclusion in preventive measures, such as vaccination, as well as increased medical care after being affected. We propose to screen mpox cases for VZV or HIV coinfection. Future data are needed to determine a possible correlation between these infections and to take necessary measures in these patient groups.

## 6. Conclusions

Our meta-analysis study highlights important differences in the epidemiologic, demographic, and clinical features of mpox cases before and after the 2022 outbreak. In the current outbreak, the possibility of changes in the pathogenic properties of the virus has been speculated because of the sudden increase in patients, especially in non-endemic areas. Therefore, our study could serve as a basis for current investigations to identify the different aspects of previous mpox outbreaks and compare them with the current ones. According to our analysis, it seems that post-2022 mpox cases experience a milder disease with fewer rashes and lower mortality rates. In addition, more than 95% of post-2022 cases have required outpatient management.

In the current global emergency, first-line medical practitioners, as well as public health policy-makers, should be aware of the previous, and recently identified, characteristics of the disease, especially clinical manifestations and epidemiological features, to make appropriate decisions and actions to control the outbreak.

## Figures and Tables

**Figure 1 biomedicines-11-00957-f001:**
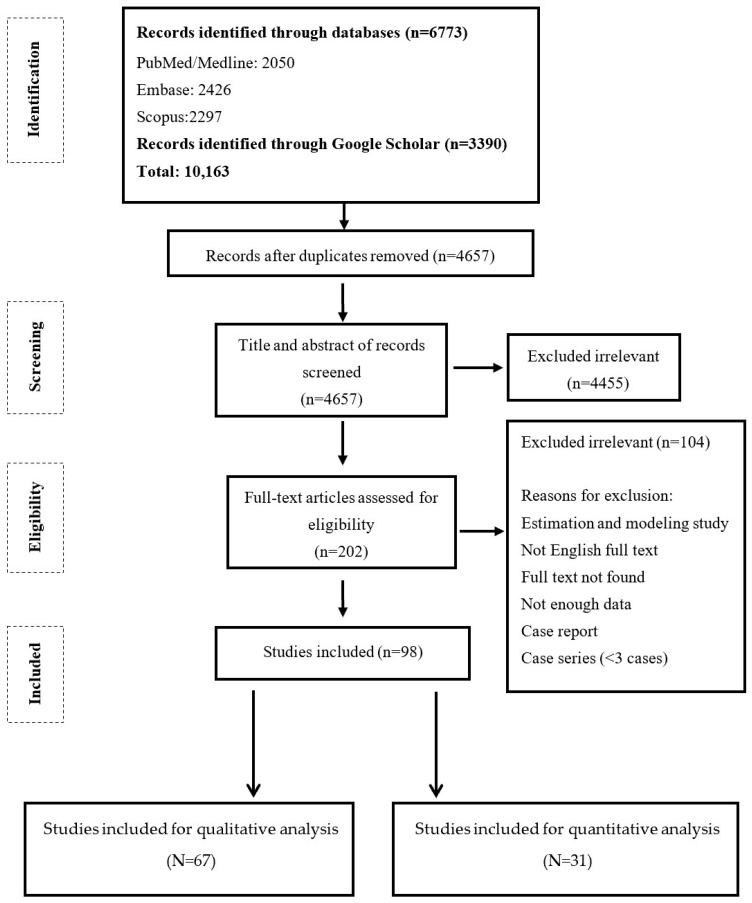
Flow chart of study selection for inclusion in the systematic review and meta-analysis.

**Figure 2 biomedicines-11-00957-f002:**
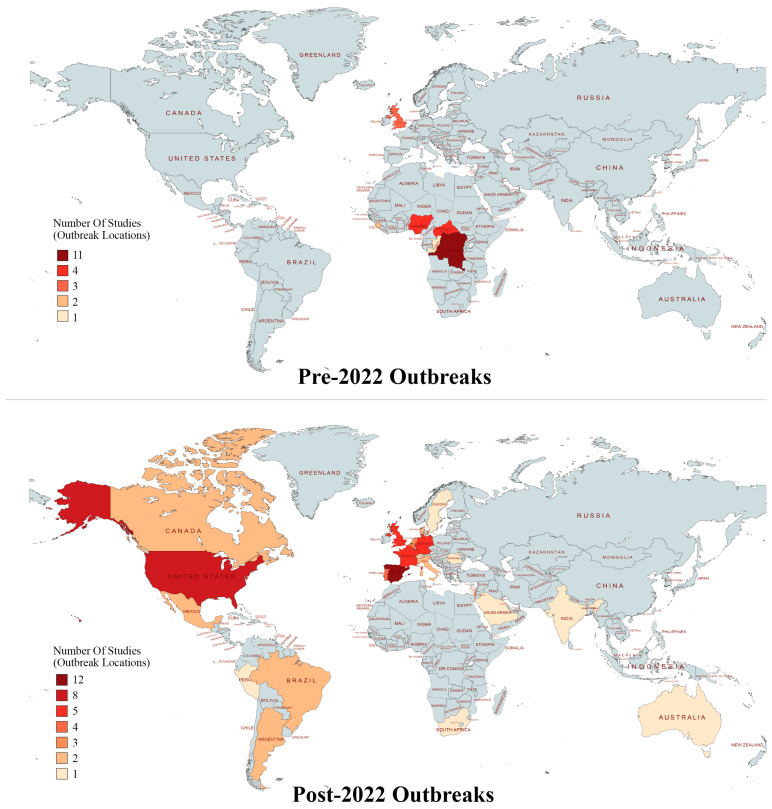
Geographical locations of the pre- and post-2022 mpox outbreak studies. The mapchart.net website was used for drawing the world map.

**Figure 3 biomedicines-11-00957-f003:**
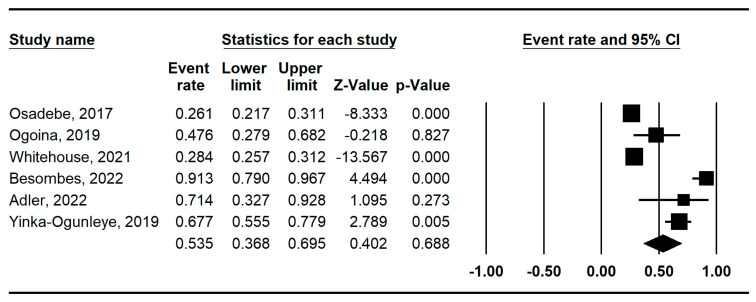
Pooled frequency of genitalia involvement in pre-2022 studies. Osadebe, 2017 [17], Ogoina, 2019 [21], Whitehouse, 2021 [23], Besombes, 2022 [24], Adler, 2022 [26], Yinka-Ogunleye, 2019 [51].

**Figure 4 biomedicines-11-00957-f004:**
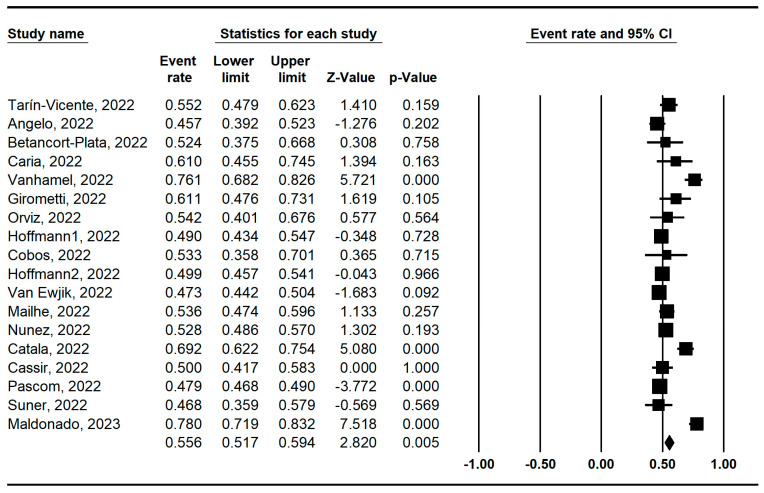
Pooled frequency of genitalia involvement in post-2022 studies. Tarín-Vicente, 2022 [27], Angelo, 2022 [28], Betancort-Plata, 2022 [29], Caria, 2022 [30], Vanhamel 2022 [31], Girometti, 2022 [33], Orviz, 2022 [34], Hoffmann1, 2022 [35], Cobos, 2022 [36], Hoffmann2, 2022 [37], Van Ewijk, 2022 [38], Mailhe, 2022 [39], Nunez, 2022 [40], Catala, 2022 [41], Cassir, 2022 [42], Pascom, 2022 [70], Suner, 2022 [43], Maldonado, 2023 [44].

**Figure 5 biomedicines-11-00957-f005:**
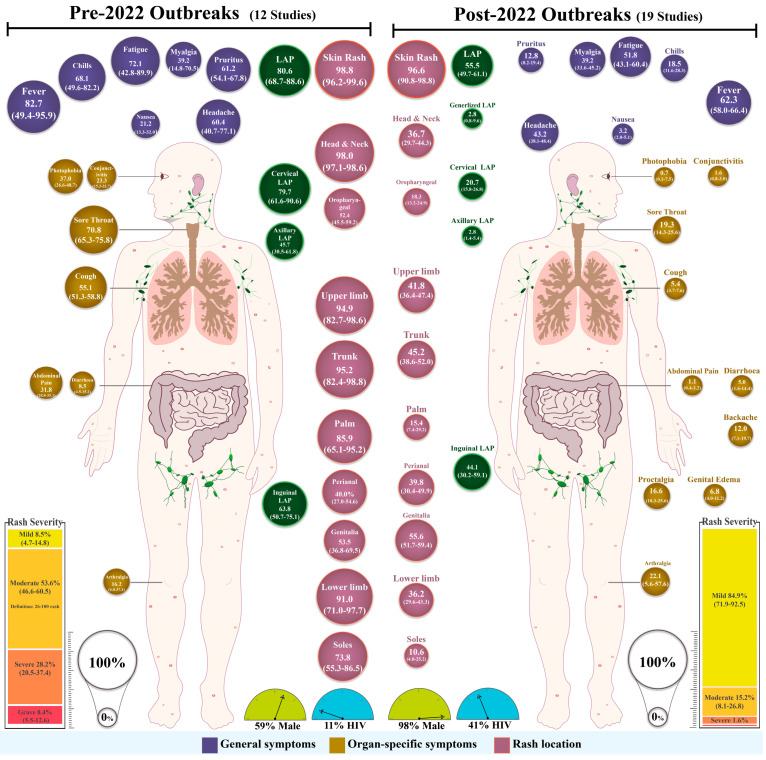
Clinical characteristics of mpox-confirmed patients adjusted with the pooled frequency.

**Table 1 biomedicines-11-00957-t001:** Quality assessment of the cross-sectional studies included in the meta-analysis (the JBI tool).

Author	Was the Sample Frame Appropriate to Address the Target Population?	Were Study Participants Sampled in an Appropriate Way?	Was the Sample Size Adequate?	Were the Study Subjects and the Setting Described in Detail?	Was the Data Analysis Conducted with Sufficient Coverage of the Identified Sample?	Were Valid Methods Used for the Identification of the Condition?	Was the Condition Measured in a Standard, Reliable Way for All Participants?	Was There Appropriate Statistical Analysis?	Was the Response Rate Adequate, and if Not, Was the Low Response Rate Managed Appropriately?	Final Appraisal
Nolen et al. [15]	Yes	Yes	No	Yes	Yes	Yes	Yes	Yes	Unclear	Included
Hoff et al. [16]	Yes	Yes	Yes	Yes	Yes	Yes	Yes	Yes	Yes	Included
Osadebe et al. [17]	Yes	Yes	Yes	No	Yes	Yes	Yes	Yes	Yes	Included
Doshi et al. [18]	Yes	Yes	Yes	Yes	Yes	Yes	Yes	Yes	Yes	Included
Doshi et al. [19]	No	No	No	Yes	No	Yes	Yes	No	Unclear	Included
Yinka-Ogunleye et al. [20]	Yes	Yes	Yes	Yes	Yes	Yes	Yes	Yes	Yes	Included
Ogoina et al. [21]	Yes	Yes	No	No	No	Yes	Yes	No	Unclear	Included
Hughes et al. [22]	Yes	Yes	Yes	Yes	Yes	Yes	Yes	No	Unclear	Included
Whitehouse et al. [23]	Yes	Yes	Yes	Yes	Yes	Yes	Yes	Yes	Yes	Included
Besombes et al. [24]	Yes	Yes	Yes	Yes	Yes	Yes	Yes	Yes	Yes	Included
Patel et al. [25]	Yes	Yes	Yes	Yes	No	Yes	Yes	Yes	Yes	Included
Adler et al. [26]	Yes	Yes	No	Yes	Yes	Yes	Yes	Yes	Yes	Included
Tarín-Vicente et al. [27]	Yes	Yes	Yes	Yes	Yes	Yes	Yes	Yes	Yes	Included
Angelo et al. [28]	Yes	Yes	Yes	Yes	Yes	Yes	Yes	Yes	Yes	Included
Betancort-Plata et al. [29]	Yes	Yes	No	No	Yes	Yes	Yes	Yes	Yes	Included
Caria et al. [30]	Yes	Yes	No	Yes	Yes	Yes	Yes	Yes	Yes	Included
Vanhamel et al. [31]	Yes	Yes	Yes	Yes	No	Yes	Yes	Yes	Yes	Included
Fink et al. [32]	Yes	Yes	Yes	Yes	Yes	Yes	Yes	Yes	Yes	Included
Girometti et al. [33]	No	Yes	No	Yes	No	Yes	Yes	Yes	Yes	Included
Orviz et al. [34]	Yes	Yes	No	Yes	Yes	Yes	Yes	Yes	Yes	Included
Hoffman et al. [35]	Yes	Yes	Yes	Yes	No	Yes	Yes	Yes	Yes	Included
Cobos et al. [36]	Yes	Yes	No	Yes	Yes	Yes	Yes	Yes	Yes	Included
Hoffmann et al. [37]	Yes	Yes	Yes	Yes	No	Yes	Yes	Yes	Yes	Included
van Ewijk et al. [38]	Yes	Yes	Yes	Yes	Yes	Yes	Yes	Yes	Yes	Included
Mailhe et al. [39]	Yes	Yes	Yes	Yes	Yes	Yes	Yes	Yes	Yes	Included
Núñez et al. [40]	Yes	Yes	Yes	Yes	Yes	Yes	Yes	Yes	Yes	Included
Catala et al. [41]	Yes	Yes	Yes	Yes	Yes	Yes	Yes	Yes	Yes	Included
Cassir et al. [42]	Yes	No	Yes	Yes	Yes	Yes	Yes	Yes	Yes	Included
Suner et al. [43]	Yes	Yes	Yes	Yes	Yes	Yes	Yes	Yes	Yes	Included
Maldonado et al. [44]	Yes	Yes	Yes	Yes	Yes	Yes	Yes	Yes	Yes	Included

JBI: the Joanna Briggs Institute.

**Table 2 biomedicines-11-00957-t002:** Characteristics of the included studies.

Authors	Year	Country	Type of Study	Cohort/Outbreak Time	Cohort/Outbreak Location	Study Population							Clade
						Total	Confirmed	Probable	Suspected	Primary Cases	Secondary Cases	Unknown	
Johnston et al. [45]	2014	USA	Short communication	2005–2007	DRC	19	19	0	0	0	0	19	NA
McCollum et al. [46]	2015	USA	Case series	November 2011	Kivu, DRC	6	3	0	3	0	0	6	NA
Nolen et al. [15]	2016	DRC	Cross-sectional	December 2013	Bokungu, DRC	63	20	19	24	0	0	63	NA
Reynolds et al. [47]	2016	USA	Brief report	2013	DRC	7	2	0	5	0	0	7	NA
Hoff et al. [16]	2017	USA	Cross-sectional	2006	DRC	1158	785 (633 MPX only + 152 MPX/VZV)	0	373	0	0	1158	NA
Mbala et al. [48]	2017	USA	Brief report	March 2007	DRC	4	4	0	0	0	0	4	NA
Osadebe et al. [17]	2017	USA	Cross-sectional	2009–2014	DRC	752	333	0	419	0	0	333	NA
Nakoune et al. [49]	2017	The Central African Republic	Brief report	December 2015–January 2016	Mbomou province, The Central African Republic	10	3	7	0	1	2	7	Zaire genotype strain
Kalthan et al. [50]	2018	The Central African Republic	Cross-sectional	August 2016–October 2016	Africa	26	3	0	23	0	23	3	NA
Yinka-Ogunleye et al. [51]	2018	Nigeria	Letter	November 2017	Nigeria	42	42	0	0	0	0	42	West African
Vaughan et al. [52]	2018	UK	Rapid communication	September 2018	UK	2	2	0	0	2	0	0	West African (Nigerian)
Doshi et al. [18]	2019	USA	Cross-sectional	November 2005–November 2007	Sankuru Province, DRC	223	223	0	0	223	0	0	NA
Raynolds et al. [53]	2019	USA	Letter	March 2014, March 2017	Kpetema town and Pujehun district, Sierra Leone	2	2	0	0	1	0	1	West African
Doshi et al. [19]	2019	USA	Cross-sectional	January 2017–April 2017	3 Enyelle, 15 Dongou, 4 Impfondo	22	7	15	0	2	11	9	NA
Ye et al. [54]	2019	China	Letter	March 2017	Kpaku village, Pujehun district, Sierra Leone	1	1	0	0	1	0	0	West African
Yinka-Ogunleye et al. [20]	2019	Nigeria	Cross-sectional	2017–2018	Nigeria	122	118	4	0	86	36	0	West African
Besombes et al. [55]	2019	France	Letter	September 2018, October 2018	Lobaya, Central African Republic	6	6	0	0	1	5	0	NA
Ogoina et al. [21]	2019	Nigeria	Cross-sectional	September 2022	Bayelsa state, Nigeria	21	18	3	0	0	0	21	West African
Hughes et al. [22]	2020	USA	Cross-sectional	September 2009	DRC	1271	534 (400 MPX only + 134 MPX/VZV)	0	737	0	0	1271	NA
Ogoina et al. [56]	2020	Nigeria	Brief report	September 2017–December 2018	Nigeria	40	40	0	0	0	0	40	NA
Whitehouse et al. [23]	2021	USA	Cross-sectional	2011–2015	DRC	1057	1057	0	0	309	279	469	NA
Hobson et al. [57]	2021	UK	Rapid communication	May 2021	UK	3	3	0	0	0	2	1	West African
Ng et al. [58]	2022	Singapore	Correspondence	April 2019	Singapore	1	1	0	0	1	0	0	NA
Besombes et al. [24]	2022	France	Cross-sectional	2001–2021	Central African Republic	327	99	0	61 suspected, 167 contacts	0	44	55	NA
Pittman et al. [59]	2022	USA	Cross-sectional	March 2007–August 2011	Democratic Republic of the Congo	216	216	0	0	0	82	134	NA
Adler et al. [26]	2022	UK	Cross-sectional	August 2018–September 2021	UK	7	7	0	0	0	3	4	NA
Tarín-Vicente et al. [27]	2022	Spain	Cross-sectional	2022	Spain	181	181	0	0	0	181	0	West African
Vallejo-Plaza et al. [60]	2022	Spain	Rapid communication	26 April 2022–21 November 2022	Spain	7393	7393	0	0	0	4986	2407	NA
Thornhill et al. [61]	2022	UK	Case series	27 April 2022–24 June 2022	United States, Canada, Mexico, Argentina, Netherlands, Belgium, United Kingdom, Portugal, Spain, France, Switzerland, Italy, Germany, Denmark, Australia, Israel	528	528	0	0	0	511	17	NA
Perez-Duque et al. [62]	2022	Portugal	Rapid communication	29 April–23 May 2022	Portugal	27	27	0	0	0	0	27	West African
Angelo et al. [28]	2022	USA	Cross-sectional	May 2022–July 2022	19 countries (Spain (35% of 226 patients), Canada (29% of 226 patients), Germany, France, Belgium, Netherlands, Portugal, Sweden, Romania, USA, Israel, South Africa, UK, Denmark, Argentina)	226	226	0	0	0	78	117	NA
Betancort-Plata et al. [29]	2022	Spain	Cross-sectional	May 2022–July 2022	Spain	42	42	0	0	0	1	41	NA
Rodríguez-Cuadrado et al. [63]	2022	Spain	Case series	May 2022–July 2022	Spain	20	20	0	0	0	0	20	NA
Caria et al. [30]	2022	Portugal	Cross-sectional	May–July 2022	Portugal	41	41	0	0	0	37	4	NA
Vanhamel et al. [31]	2022	Belgium	Cross-sectional	May 2022–Jun 2022	Belgium	139	139	0	0	66	28	45	NA
Fink et al. [32]	2022	UK	Cross-sectional	6 May–August 2022	UK	156	156	0	0	0	0	156	NA
Thornhill et al. [64]	2022	UK	Case series	11 May 2022–4 October 2022	15 countries and three WHO regions; 65 regions of America, 68 European regions, 3 African regions	136	126	10	0	0	118	18	NA
Patel et al. [25]	2022	UK	Case series	13 May 2022–1 July 2022	UK	197	197	0	0	0	41	156	NA
Girometti et al. [33]	2022	UK	Cross-sectional	14–25 May 2022	UK	54	54	0	0	0	2	52	NA
Vivancos-Gallego et al. [65]	2022	Spain	Brief report	16 May 2022–Jun 2022	Spain	25	25	0	0	0	0	25	West African
Sheffer et al. [66]	2022	Israel	Short communication	16 May 2022–13 September 2022	Israel	203	203	0	0	0	198	5	NA
Antinori et al. [67]	2022	Italy	Case series	17–22 May 2022	Italy	4	4	0	0	0	0	4	NA
Orviz et al. [34]	2022	Spain	Cross-sectional	18 May 2022–first weeks of June 2022	Spain	48	48	0	0	0	7	41	West African
Hoffmann et al. [35]	2022	Germany	Cross-sectional	19 May 2022–June 2022	Germany	301	301	0	0	0	0	301	NA
Cobos et al. [36]	2022	Spain	Cross-sectional	19 May–7 June 2022	Spain	30	30	0	0	0	30	0	NA
Hoffmann et al. [37]	2022	Germany	Cross-sectional	19 May 2022–30 June 2022	Germany	546	546	0	0	0	0	546	NA
Van Ewijk et al. [38]	2022	Netherlands	Cross-sectional	20 May 2022–8 August 2022	Netherlands	1928	1000	122	806	0	865	135	NA
Mailhe et al. [39]	2022	France	Cross-sectional	21 May 2022–5 July 2022	France	264	264	0	0	0	112	152	NA
Núñez et al. [40]	2022	Mexico	Cross-sectional	24 May 2022–5 September 2022	Mexico	565	565	0	0	0	104	461	West African
Maldonado-Barrueco et al. [68]	2023	Spain	Case series	26 May 2022–31 December 2022	Spain	30	30	0	0	0	0	30	NA
Catala et al. [41]	2022	Spain	Cross-sectional	28 May–14 July 2022	Spain	185	185	0	0	0	67	118	NA
Relhan et al. [69]	2022	India	Short communication	Jun 2022–Aug 2022	India	5	5	0	0	0	1	4	NA
Cassir et al. [42]	2022	France	Cross-sectional	4 June 2022–31 August 2022	France	136	136	0	0	0	28	108	NA
Pascom et al. [70]	2022	Brazil	Cross-sectional	7 Jun 2022–1 Oct 2022	Brazil	8176	7992	175	0	0	0	8176	NA
Martins-Filho et al. [71]	2022	Brazil	Letter	9 June 2022–23 November 2022	Brazil	9100	9100	0	0	0	0	1457	NA
Wong et al. [72]	2022	USA	Case seies	13 Jun 2022–11 Jul 2022	USA	7	7	0	0	0	7	0	NA
Suner et al. [43]	2022	Spain	Cross-sectional	28 June 2022, and 22 September 2022	Spain	77	77	0	0	0	0	77	NA
Cash-Goldwasser et al. [73]	2022	USA	Case series	July–September 2022	USA	5	5	0	0	0	0	5	NA
Ciccarese et al. [74]	2022	Italy	Case series	1 July–31 August 2022	Italy	16	16	0	0	0	16	0	NA
Aguilera-Alonso et al. [75]	2022	Spain	Correspondence	Until August 2022	Spain	16	16	0	0	0	15	1	NA
Gnanaprakasam et al. [76]	2022	USA	Letter	Summer 2022	USA	23	23	0	0	0	0	23	NA
Choudhury et al. [77]	2022	Germany	Letter	Until September 2022	Germany	179	179	0	0	0	150	29	NA
Srichawla et al. [78]	2022	USA	Case series	NA	USA	9	9	0	0	0	7	2	West African
Maldonado et al. [44]	2023	Peru	Cross-sectional	1 July 2022–3 September 2022	Peru	205	205	0	0	0	17	188	NA
Rekik et al. [79]	2023	France	Case series	6–11 July 2022	France	20	20	0	0	0	9	11	NA
Prasad et al. [80]	2023	USA	Case series	4 August 2022–13 November 2022	13 countries in North America, Europe, Asia, Latin America and the Caribbean, and Africa	101	101	0	0	0	32	69	NA
Assiri et al. [81]	2023	Saudi Arabia	Case series	NA	Saudi Arabia	7	7	0	0	0	7	0	NA

NA: not available.

**Table 3 biomedicines-11-00957-t003:** Confirmed, probable, and suspected case definitions of the included studies (only available data are mentioned), compared to WHO and CDC case definitions.

First Author	Confirmed	Probable	Suspected
Nolen et al. [15]	(PCR of Orthopoxvirus **OR** DNA signature of MPOX) **AND** fever/rash **AND** 1/3 criteria *	Fever **AND** rash **AND** 1/3 criteria * **AND** contact in 14d	Fever **AND** Rash **AND** 1/3 criteria *
Reynolds et al. [47]	PCR assay	NA	Vesicular pustular eruption characterized by a hard and deep pustule **AND** 1/3 criteria *
Hoff et al. [16]	qPCR	NA	Preceding fever **AND** rash **AND** (face/palms/soles **OR** >5 smallpox-type scabs)
Mbala et al. [48]	WHO (pan-orthopoxvirus MGB-hemagglutinin real-time PCR assay)	WHO criteria	WHO criteria
Osadebe et al. [17]	MPOXV-specific RT-PCR **OR** OPXV-specific assay	NA	Plan A (for endemic situations): preceding fever **AND** rash **AND** (face/palms/soles **OR** >5 smallpox type scabs); Plan B (discriminate VZV and MPOX): vesicular or pustular eruption with deep-seated **AND** 1/3 criteria *
Kalthan et al. [50]	Suspected case **AND** (PCR **OR** virus isolation on baby mouse brain cell culture	NA	Living in the district of Alindao **AND** fever **AND** rash
Doshi et al. [18]	PCR Only	NA	Sudden preceding fever **AND** rash
Doshi et al. [19]	OPXV DNA detection by PCR	One of the epidemiologic criteria *** **AND** demonstrated elevated levels of OPXV-specific IgM **AND** having an unexplained rash and fever and >2 other signs or symptoms from the clinical criteria	Unexplained rash and fever
Yinka-Ogunleye et al. [20]	Suspected case **AND** (viral identification by RT-PCR **OR** virus isolation **OR** antibody detection)	Suspected **AND** link with a known case **AND** no access to laboratory test	Sudden preceding fever **AND** rash on the face, palms, and soles
Ogoina et al. [21]	Suspected **AND** (positive IgM antibody and PCR **OR** virus isolation)	Suspected **AND** link with a known case **AND** no access to the laboratory	Sudden preceding fever **AND** rash on the face, palms, and soles
Hughes et al. [22]	MPOXV-specific RT-PCR **OR** OPXV-specific assay	NA	Vesicular pustular eruption characterized by a hard and deep pustule **AND** 1/3 criteria *
Ogoina et al. [56]	Suspected **AND** (positive IgM antibody and PCR **OR** virus isolation)	Suspected **AND** link with a known case **AND** no access to the laboratory	Sudden preceding fever **AND** rash on the face, palms, and soles
Whitehouse et al. [23]	RT-PCR **OR** virus isolation	NA	Vesicular and pustular eruptions characterized by a hard and deep pustule **AND** 1/3 criteria *
CDC ** [82]	Monkeypox virus DNA by PCR by next-generation sequencing **OR** isolation of Monkeypox virus in culture	No suspicion of other recent OPXV exposure **AND** (OPXV DNA **OR** immunohistochemical **OR** electron microscopy **OR** detectable levels of anti-OPXV IgM antibody during the period of 4 to 56 days after rash onset)	New characteristic rash **OR** one epidemiologic criterion ***
WHO (21 May 2022) [83]	Viral DNA by RT-PCR **OR** sequencing	Suspected case **AND** (epidemiological link **OR** multiple or anonymous sexual partners in the 21 days before symptom onset **OR** anti-OPXV IgM 4–56 days before rash onset **OR** four-fold rise in IgG Ab 5–7 days before rash onset and convalescent on day 21 onwards **OR** known exposure to OPXV **OR** OPXV-specific PCR without MPOXV-specific PCR or sequencing)	Unexplained acute rash/ skin lesion **AND** cannot be explained by other causes of skin lesions **AND** (headache **OR** acute onset fever **OR** lymphadenopathy **OR** myalgia **OR** back pain **OR** asthenia)
ECDC [84]	MPOXV-specific PCR assay **OR** OPXV-specific PCR assay positive confirmed by MPOXV sequencing with symptoms	Unexplained generalized or localized maculopapular or vesiculopustular rash with centrifugal spread, with lesions showing umbilication or scabbing, lymphadenopathy, and one or more other MPOX-compatible symptoms **OR** unexplained rash **AND** (positive OPXV PCR **OR** epidemiological link **OR** travel to MPOX endemic countries **OR** multiple or anonymous sexual partners **OR** man who has sex with men)	NA
India CDC [85]	Viral DNA by RT-PCR **OR** sequencing	Suspected case definition **AND** epidemiological link with a confirmed case **OR** a compatible case	Unexplained acute rash **AND** (swollen lymph nodes **OR** fever **OR** headache **OR** body aches **OR** profound weakness)

PCR: polymerase chain reaction; RT-PCR: real-time PCR; MPOXV: human monkeypox virus; OPXV: orthopoxvirus; CDC: center for disease control and prevention; ECDC: European CDC; NA: not available; Ab: antibody * 1/3 criteria: one of three criteria of fever preceding the eruption, lymphadenopathy, pustules or crusts on the palms of the hands or soles of the feet; ** CDC exclusion criteria: alternative diagnosis can fully explain the illness OR an individual with symptoms consistent with monkeypox does not develop a rash within 5 days of illness onset OR a case where high-quality specimens do not demonstrate the presence of Orthopoxvirus or Monkeypox virus or antibodies to orthopoxvirus; *** CDC Epidemiologic criteria (within 21 days of illness onset) exposure with probable case/similar appearing rash OR traveled outside the US to a country with confirmed cases of monkeypox or where Monkeypox virus is endemic OR had contact with a dead or live wild animal or exotic pet that is an African endemic species or used a product derived from such animals OR intimate in-person contact with individuals in a social network experiencing monkeypox activity, and this includes men who have sex with men.

**Table 4 biomedicines-11-00957-t004:** Demographic and epidemiologic characteristics of the study population (confirmed cases).

Authors	Mean Age	Gender	Nationality	Epidemiological History and Associated Risk Factors	Diagnostic Confirmation Tool
Johnston et al. [45]	11.7	7M, 12F	NA	NA	NA
McCollum et al. [46]	25	2M, 1F	African	Traveling from a highly forested area, animal contact (monkey), eating monkey meat, farmer occupation	PCR (vesicular swab, crust, blood)
Nolen et al. [15]	20.4	12M, 8F	DRC	NA	NA
Reynolds et al. [47]	11.5	2F	African	Human contact	PCR (vesicular swab)
Hoff et al. [16]	<5 years: 193 (153MPX only + 40MPX/VZV) 5–14 years: 353 (297MPX only + 56MPX/VZV) 15–29 years: 198 (154MPX only + 44MPX/VZV) >30 years: 41 (29MPX only + 12MPX/VZV)	480M (388MPX only + 92MPX/VZV), 305F (245MPX only + 60MPX/VZV)	NA	NA	PCR (vesicular fluid or crust)
Mbala et al. [48]	24	4F	NA	NA	PCR
Osadebe et al. [17]	5.77	178M, 154F	Congo	NA	NA
Nakoune et al. [49]	17	2M, 1F	Central African	Animal contact, human contact	Viral DNA minikit (Qubit dsDNA BR Assay kit), PCR
Yinka-Ogunleye et al. [51]	30	28M, 14F	Nigeria	Animal contact	PCR, IgG, sequencing
Vaughan et al. [52]	Adult	NA	1 UK, 1 Nigerian	Traveling from Africa, human contact	PCR, molecular assay
Doshi et al. [18]	12	155M, 68F	DRC	Animal contact (eating, cooking, butchering/skinning, and hunting rodents and non-human primates), visiting a forest Occupation: 134 students, 26 farmers	PCR
Raynolds et al. [53]	17.95	2M	Sierra Leone	Animal contact (rodents), consuming wild animal meat, farmer occupation	PCR, IgG, IgM
Doshi et al. [19]	17.71	2M, 5F	DRC	Hunter occupation	PCR, IgM
Ye et al. [54]	35	1M	Sierra Leone	Traveling from Pelewahun gee bu, animal contact (hunting and eating squirrels)	PCR, IgG, sequencing
Yinka-Ogunleye et al. * [20]	29	84M, 38F	Nigerian	Human contact (living in the same household or prison with patients, healthcare occupation)	PCR, IgM
Besombes et al. [55]	17	6F	Central African Republic	Animal contact (butchering small mammals), human contact	PCR
Ogoina et al. * [21]	29	17M, 4F	Nigerian	NA	RT-PCR of blood, swab, or crust (at least 2 specimens), serology, culture
Hughes et al. [22]	15.7 (MPX only: 15.9, MPX/VZV: 15.5)	279M (212MPX only + 65MPX/VZV), 255F (188MPX only+ 67MPX/VZV)	NA	NA	PCR
Ogoina et al. [56]	28	31M, 9F	Nigerian	NA	NM
Whitehouse et al. [23]	14	568/1054M, 486/1054F	Congo	Animal contact, human contact	PCR, QIAamp DNA Blood Mini Kit (Qiagen)
Hobson et al. [57]	NA	NA	NA	Traveling from Nigeria, household transmission	PCR (vesicular swab)
Ng et al. [58]	38	1M	Nigerian	Traveling from Africa	PCR, electron microscopy, genome sequencing of blister fluid
Besombes et al. [24]	15.5 (median)	45M, 51F, 3 missing data	NA	Human contact: 44/99 reported contact with a human case Transmission route: 504/528 sexual close contact, 4/528 nonsexual close contact, 17/528 other or unknown Occupation: 16/99 farmer, 6/99 hunter/fisherman, 0/99 healthcare worker, 2/99 mine worker, 2/99 market trader, 9/99 other, 27/99 missing data MSM: 509/528	RT-PCR
Pittman et al. [59]	14	138M, 78F	NA	Animal contact: 133/216 handled uncooked, freshly butchered meat, 88/216 meat of ground squirrel, 82/216 meat of monkey, 47/216 dead animal, 11/216 meat of Gambian rat or other rodents, 156/216 clean/dressed consumption of wild game, 46/216 other wild game Human contact: 86/216 household, 57/216 initial Mpx contact with blood, body fluid, or person with tissue or secretion	RT-PCR
Adler et al. [26]	NA	4M, 3F	NA	Travel: 6/7 travel history Human contact: 3/7 human contact Occupation: 1/7 healthcare worker	PCR
Tarín-Vicente et al. [27]	37 (median)	175M, 6F	79 Spanish, 82 south and central American, 19 other, 1 missing data	Travel: 26/181 travel out of Spain, 0/181 travel to endemic regions Human contact: 47/181 regular sexual partner with monkeypox, 6/181 household contact with monkeypox, 66/181 attendance at a Pride event MSM: 166/181 Mentioned behavioral risk factors: number of sexual partners in past 14 days: 2 (median), number of sexual partners in past 3 months: 6.5 (median), 99 sexually transmitted infections in the past 12 months, 107 use of social media apps to identify sexual partners, 15 sex outside of Spain in past 3 months, 8 sex with a sex worker, 57 use of recreational drugs during sex, 11 vaginal-insertive sex, 6 vaginal-receptive sex, 131 anal-insertive sex, 108 anal-receptive sex, 160 oral-insertive sex, 158 oral-receptive sex	RT-PCR
Vallejo-Plaza et al. [60]	In women: 34 (median) In men: 37 (median)	7235M, 158F	NA	Transmission route: 4639/5023 sexual transmission (women 69/105. men 4570/4918) human to human, non-sexual: 347/5023 (women: 23/105, men 324/4918) other, not specified: 37/5023 (women 13/105, men 24/4918)	RT-PCR
Thornhill et al. [61]	38 (median)	527M	Race: 398 White, 25 Black, 19 mixed, 66 Latinx, 20 other or unknown	Travel: 147/528 travel abroad in the month before diagnosis Human contact: 135/528 contact with a person known to have monkeypox Transmission route: 504/528 sexual close contact, 4/528 nonsexual close contact, 17/528 other or unknown, 3/528 household contact MSM: 509/528 Mentioned behavioral risk factors: 5 median number of sex partners in 406 patients in the previous 3 months, 169/406 visited sex-on-site venues within the previous month, 106/406 engaged in chemsex.	RT-PCR
Perez-Duque et al. [62]	34.1	27M	NA	Travel: 4/27 Animal contact: 3/23 Human contact: 1/10 MSM: 18/19 Mentioned behavioral risk factors: 14/16 multiple sex partners, 6/27 attendance at the sauna	RT-PCR and/or nucleotide sequencing
Angelo et al. [28]	37 (median)	226M	NA	Travel: 37/210 international travel in the 21 days before symptom onset (the most frequently reported travel destinations and reasons were European countries for tourism (30 [83%] of 36 trips)). Animal contact: 10/134 touched any live animals in the 21 days before symptom onset (all contact was with domesticated cats or dogs), 3/140 touched any dead animals in the 21 days before symptom onset (exposures included butchering, handling, or cooking meat from wild animals (n = 2) and eating animal products from a store (n = 1)). Human contact: 78/195 known close contact with a suspect or confirmed human monkeypox case (type of contact: 70/71 sexual or close intimate contact, 8/71 household contact, 2/71 face-to-face contact not in the household, 3/71 other) Occupation: 8/168 healthcare workers (all eight were MSM and four (50%) met sexual partners at a mass gathering; there was no evidence of nosocomial transmission.) MSM: 208/211 Mentioned behavioral risk factors: 37/161 met their sexual partners at mass gatherings, including the Maspalomas Festival in Spain, and various other Pride-related festivities in Europe and the USA, 216/219 reported sexual or close intimate contact in the 21 days before illness onset, 1/169 lived in congregate setting	PCR
Betancort-Plata et al. [29]	40 (median)	42M	18/42 Spanish, 3/42 Germany, 1/42 France, 1/42 United Kingdom, 3/42 Italy, 3/42 Poland, 1/42 Portugal, 1/42 Russia, 1/42 Switzerland, 1/42 Argentina, 1/42 Brazil, 1/42 Colombia, 3/42 Cuba, 1/42 Honduras, 1/42 Peru, 1/42 Dominican Republic, 1/42 Morocco	Human contact: 37/42 recent sexual contact with other men, 2/42 contact with women, 1/1 contact with a person with a skin lesion MSM: 37/42	RT-PCR
Rodríguez-Cuadrado et al. [63]	40.5	20M	NA	NA	RT-PCR
Caria et al. [30]	37.2	40M, 1F	18/41 Brazilian, 15/41 Portuguese, 2/41 French, 2/41 Colombian, 1/41 Spanish, 1/41 Peruvian, 1/41 Cape Verdean, 1/41 Lebanese	Travel: 7/41 recent international travel to countries in Europe (1/41 Germany, 1/41 Spain, 1/41 Belgium, and 1/41 the UK), Asia (2/41 Israel), and the Americas (1/41 Dominican Republic) in the month preceding symptom onset Human contact: 16/41 sex contact with MPX-confirmed cases MSM: 38/41 Mentioned behavioral risk factors: 37/41 sex with multiple and/or anonymous partners or unprotected sex in the previous month, 16/41 sex with MPX-confirmed case, 6/41 sex party or venue attendance in the previous month, 8/41 “Chemsex” in the previous month	Laboratory-confirmation
Vanhamel et al. [31]	38 (median)	139M	NA	Travel: 52/139 travel outside Belgium in the 21 days before symptom onset Human contact: 119/139 Transmission route: 115/139 sexual contact, 4/139 other person-to-person transmission MSM: 113/136	PCR
Fink et al. [32]	35 (median)	153M, 3F	Race: 105/156 White, 12/156 Black, 11/156 Latinx, 6/156 South Asian, 14/156 other, 8/156 unknown	MSM: 139/155	PCR
Thornhill et al. * [64]	34 (median)	136F (69 cis women, 62 trans women, 5 non-binary individuals assigned female at birth)	Race: 61 Latinx, 40 White, 28 Black, 3 Asian, 2 mixed, 1 First nation, Inuit, or Métis 1 other or unknown	Human contact: 38/136 known contact with the monkeypox virus Occupation: 35/136 sex work, 13/136 health care, 11/136 business or office work, 7/136 student, 3/136 cleaning, 4/136 food service, 2/136 beauty, 5/136 sales or marketing, 1/136 arts, 1/136 teaching or childcare, 6/136 other, 19/136 unemployed, 29/136 unknown Transmission route: 4/136 occupational exposure (health-care workers), 7/136 household, 7/136 non-sexual close contact, 100/136 sexual contact, 18/136 unknown Mentioned behavioral risk factors: 14/136 injecting drugs, 36/136 current sex work, 9/136 attended Pride events or similar within the month preceding symptom onset, 10/136 attended large event or festival within the month preceding symptom onset, 8/136 homeless	RT-PCR
Patel et al. [25]	38 (median)	197M	NA	Travel: 54/197 participants had a history of travel abroad within four weeks before symptom onset. The most common destinations were within western Europe: Spain (20), France (8), Belgium (4), Germany (4), and Greece (4). One participant had returned from an endemic area (West Africa). Human contact: 41/155 known close contact with someone who showed symptoms of or had confirmed monkeypox infection MSM: 196/197 Mentioned behavioral risk factors: 170/177 reported sexual contact with a male partner within 21 days of symptoms developing	RT-PCR
Girometti et al. [33]	41 (median)	54M	26/54 born in the UK, 38/54 White, 8/54 Black or mixed race, 4/54 Asian, 4/54 other ethnicities	Travel: 25/54 a history of travel outside the UK within the previous 2 months (eight to Spain, five to France, and two to the Netherlands; none reported travel to sub-Saharan Africa.) Human contact: 2 contacts of a confirmed monkeypox virus case MSM: 54/54 Mentioned behavioral risk factors: 29/52 reported more than five sexual partners in the 12 weeks before the monkeypox virus diagnosis, 18/52 individuals reported more than ten sexual partners in the 12 weeks before the monkeypox virus diagnosis, 49/52 individuals reported inconsistent condom use in the 3 weeks before symptom onset, 47/52 reported at least one new sexual partner during the same period.	RT-PCR
Vivancos-Gallego et al. [65]	39.5 (median)	25M	NA	MSM: 25/25	RT-PCR
Sheffer et al. [66]	35 (median)	203M	NA	Travel: 61/203 trips outside of Israel, mostly in Europe Human contact: 195/205 reported having sexual contact as the source of infection in the 3 weeks before disease onset Occupation: 1 physician working in the emergency service who had provided medical care for patients with MPX Transmission route: 197 sexual contact/ close physical contact, 1 nosocomial infection, 5 unknown MSM: 195/203 Mentioned behavioral risk factors: 110/203 had more than three sexual partners	RT-PCR
Antinori et al. [67]	30	4M	NA	Travel: 4/4 Mentioned behavioral risk factors: 3/4 mass gathering	RT-PCR
Orviz et al. [34]	35 (median)	48M	NA	Travel: 16.7% of the patients had traveled outside of Spain three weeks before the onset of the symptoms, 0 patients had traveled or had contact with people from endemic areas Human contact: 7/48 knew other people living in the same household with similar symptoms Occupation: 2/48 homosexual sex worker MSM: 42/48 Mentioned behavioral risk factors: 89.5% of the patients had unprotected sex in the three weeks before the onset of the symptoms, 5 median number of different sexual partners per person for 21 days before the onset of the symptoms, 39/48 people practicing unprotected oral sex, 41/48 unprotected anal intercourse, 2/48 unprotected vaginal sex, 24/48 had participated in a chemsex session within the 21 days before the onset of symptoms	PCR
Hoffmann et al. [35]	39 (median)	301M	166/229 German	MSM: 301/301	PCR
Cobos et al. [36]	33	30M	Spanish 13/30, Venezuelan 9/30, Syrian 2/30, Italian 1/30, Cuban 1/30, Honduran 1/30, Peruvian 1/30, Ecuadorian 1/30, Brazilian 1/30	Travel: none had a history of travel to Central Africa in the previous 3 months MSM: 30/30 Mentioned behavioral risk factors: 30/30 unprotected sex in the 4 weeks before the clinical onset	PCR
Hoffmann et al. [37]	39 (median)	546M	313/439 German	MSM: 546/546	PCR
Van Ewijk et al. [38]	37 (median)	987M, 10F, 3 Unknown	Netherlands (511/893), Turkey (2/893), Morocco (4/893), Netherlands Antilles-Surinam and Aruba (44/893), other Western countries, including Europe, North America, Oceania, Indonesia, and Japan (159/893), and other non-Western countries (173/893), all other (107/893)	Human contact: 227 notified contact of a mpox case, 822/1000 sexual contact, 15/1000 direct unprotected contact, 5/1000 household, 20/1000 prolonged face-to-face contact, 3/1000 Occupation: 55 healthcare workers MSM: 935/987	RT-PCR
Mailhe et al. [39]	35 (median)	262M, 1F, 1 transgender woman	178/245 born in France	Travel: 76/227 Animal contact: 38/206 Human contact: 112/236 were aware of being in contact with a confirmed case of MPXV of whom 86/91 had sexual contact MSM: 245/259 Mentioned behavioral risk factors: 90/216 chemsex, 106/264 condomless sex, 5 median number of sexual partners over the last month	PCR
Núñez et al. [40]	34 (median)	549M, 16F	NA	Travel: 63/565 recent national flights, 46/565 recent international flight Occupation: 10/565 healthcare worker Transmission route: 94/565 sexual contact, 10/565 non-sexual contact, 461/565 unknown MSM: 327/335 Mentioned behavioral risk factors: 3/565 participated in group sex event	RT-PCR
Maldonado-Barrueco et al. [68]	36	30M	11/30 natives of South or Central America	MSM: 29/30	RT-PCR
Catala et al. [41]	38.7	185M	NA	Travel: 51/185 travel outside the home town or city in the 3 weeks before the first sign or symptom Animal contact: 28/185 pets in the household, 16/185 exotic pets in the household Human contact: 1/126 healthcare worker, 43/185 other contact with a case Occupation: 1/126 healthcare worker MSM: 184/185 Mentioned behavioral risk factors: 102/185 use of social networks to meet partners, 11/185 sex with sex workers in the previous 3 months, 62/185 use of drugs during sexual relationships in the previous 3 months	PCR
Relhan et al. [69]	31.2	3M, 2F	1 Indian, 4 Nigerian	Human contact: 1 Indian, 4 Nigerian Occupation: 1/5 data analyst, 1/5 chef, 1/5 businessman, 1/5 cloth factory worker, 1/5 student	RT-PCR
Cassir et al. [42]	36 (median)	133M, 3F	NA	Travel: 17/136 recent travel to an epidemic country, which includes Spain (n = 11 patients), United States (n = 2 patients), Germany (n = 1 patient), Belgium (n = 1 patient), Canada (n = 1 patient), and Italy (n = 1 patient). Human contact: 21/136 sexual partner with monkeypox MSM: 125/136 Mentioned behavioral risk factors: 7/30 attendance at a Pride event	RT-PCR
Pascom et al. * [70]	32 (median)	5881M, 542F, 6 transvestite, 70 non-binary, 104 other, 1564 not informed	Race: 3572/8176 White, 3332/8176 Black, 87/8176 Asian, 13/8176 Indigenous, 1163/8176 not informed	MSM: 4502/5189	
Martins-Filho et al. [71]	NA	8386M, 714F	NA	Human contact: 579/1457 sexual exposure before the onset of signs and symptoms	NA
Wong et al. [72]	41 (median)	7M	NA	Travel: 1/7 traveled to the Bahamas about six weeks before Human contact: 26% had known monkeypox contact MSM: 7/7	NA
Suner et al. [43]	35 (median)	75M, 1F, 1 transgender female	36 Spain, 31 latin America, 9 other EU, 1 West Africa	MSM: 70/75	RT-qPCR
Cash-Goldwasser et al. [73]	33 (median)	4M, 1F	NA	NA	PCR
Ciccarese et al. [74]	37 (median)	16M	10/16 Italian, 3/16 Ecuadorian, 1/16 Russian, 1/16 Jumaican, 1/16 Brazilian	Travel: 3/16 history of foreign travel in the month before the disease onset Human contact: 4/16 had sexual exposure to an individual known to have MPX in the week before the diagnosis, and 12/16 had risk factors for STIs, such as multiple sexual partners and/or unprotected sex in the 2 weeks before the onset of symptoms MSM: 14/16	RT-PCR
Aguilera-Alonso et al. [75]	15 (median)	10M, 6F	16/16 Spain	Transmission route: 3/16 houshold contact, 1/16 unknown, 9/16 Contact with contaminated material, 3/16 sexual close contact	RT-PCR
Gnanaprakasam et al. [76]	NA	23M	Race: 9/23 Black/African American,10/23 Hispanic, 1/23White, 1/23Unknown	Travel: 1 recent travel to a large social event for gay and bisexual men in Florida MSM: 21/23	RT-PCR
Choudhury et al. [77]	38 (median)	179M	NA	Transmission route: 137/179 sexually, 13/179 not sexually, 29/179 unknown MSM: 164/179	Laboratory-confirmation
Srichawla et al. [78]	28.4	9M	NA	Travel: 1/9 traveled to Europe in the past 30 days MSM: 9/9 Mentioned behavioral risk factors: 9/9 unprotected sexual encounter within 4 weeks of the onset of symptoms, 1/9 engaged in high-risk sexual activity involving condomless penetrative sexual intercourse with multiple men	RT-PCR
Maldonado et al. [44]	32 (median)	202M, 3F	NA	Travel: 27/205 Human contact: 17/205 MSM: 166/205 Mentioned behavioral risk factors: 179/205 sexual encounters in the past 21 days, 3 median number of sex partners in the last year, 112/205 last sexual encounter with an unknown partner, 133/205 casual sex partners, 17/205 identified sexual contact with a confirmed mpox diagnosis	RT-PCR
Rekik et al. [79]	33 (median)	20M	NA	Human contact: 9/20 past contact with a person infected with MPX Occupation: 1 transsexual sex worker MSM: 20/20 Mentioned behavioral risk factors: 8 median number of sexual partners in the last three months, 11/20 had chemsex, 18/20 anal receptive intercourse	RT-PCR
Prasad et al. [80]	35 (median)	98M, 3F	Race: 63/101White, 11/101 Black/African American, 20/101 Hispanic/Latino, 3/101Asian, 2/101 other, 3/101 missing	Human contact: 26/32 sexual contact (majority engaged in same sex sexual behavior (87%) and group sex activities (27%)) Mentioned behavioral risk factors: 87% same-sex sexual behavior, 27% sexual activity between greater than two people	RT-PCR and biopsy
Assiri et al. [81]	30.14	7M	NA	Travel: 7/7 Human contact: 2/7 heterosexual contact with a female who appeared healthy, 1/7 heterosexual contact (with a female who appeared healthy but had earlier exposure to a known MPX case), 1/7 intimate (skin-to-skin) contact with a female partner Transmission route: 4/7 acquired the Mpox virus probably through heterosexual contact with individuals who did not have obvious signs or symptoms, 3/7 seemed to have contracted Mpox virus infection through skin-to-skin contact with other individuals or indirect contact with contaminated objects in locations where outbreaks of Mpox were ongoing Mentioned behavioral risk factors: 1/7 attending music festivals and nightclubs in the community with ongoing MPX outbreaks, 2/7 body massage at unlicensed parlors	PCR

NA: not available, M: male, F: female, MSM: male having sex with males. * Data were not defined in confirmed patients and were reported in the total population; however, available data were included according to the majority number of confirmed patients among the total population.

**Table 5 biomedicines-11-00957-t005:** Clinical characteristics of the study population (confirmed cases).

Authors	Smallpox Vaccination Status	Coinfections	Clinical Manifestations	Incubation Period	Duration of Disease	Complications	Management Status	Outcome
Johnston et al. [45]	19 Unknown	NA	NA	NA	NA	NA	NA	19 Unknown
McCollum et al. [46]	3 Unknown	NA	Data of 1 case available: Pustules on the face and chest (10 days after illness onset), pustules on the palms (13 days after illness onset), scarring and hypopigmentation on the back after separation of crusts, and appearing pustules on the left foot (20 days after illness onset), fever, cough, swollen appearance of the face	7–17 days	10 days (1 case available)	NA	1 Inpatient	1 Recovered, 2 Unknown
Nolen et al. [15]	5 Yes, 15 Unknown	NA	NA	Average: 9.6 days	NA	NA	NA	20 Unknown
Reynolds et al. [47]	2 Unknown	NA	Data of 1 case available: Fever, chills, myalgia, LAP, generalized skin rash, headache, cough, dyspnea	NA	NA	NA	1 Inpatient	2 Unknown
Hoff et al. [16]	30 Yes (19MPX only + 11MPX/VZV), 755 o (614MPX only + 141MPX/VZV)	152VZV	778 fever (627MPX only + 151MPX/VZV), 209 fatigue/malaise (179MPX only + 30MPX/VZV), 687 LAP (576MPX only + 111MPX/VZV), 783 skin rash (631MPX only + 152MPX/VZV), 121 nausea/vomiting (100MPX only + 21MPX/VZV), 96 diarrhea (76MPX only + 20MPX/VZV), 141 conjunctivitis (119MPX only + 22MPX/VZV), 410 cough (337MPX only + 73MPX/VZV), 253 abdominal pain (209MPX only + 44MPX/VZV), 556 sore throat/odynophagia (467MPX only + 89MPX/VZV), 191 joint pain (159MPX only + 32MPX/VZV), 3 convulsion (2MPX only + 1MPX/VZV), 783 eruptions (631MPX only + 152MPX/VZV) **Rash location:** 468 palms (469MPX only + 99MPX/VZV), 460 soles (389MPX only + 71MPX/VZV), 384 oropharyngeal (327MPX only + 57MPX/VZV) **LAP location**: 647 cervical(543MPX ONLY + 104MPX/VZV), 433axillary (363MPX ONLY + 70MPX/VZV), 457 inguinal(387MPX ONLY + 70MPX/VZV) **Rash severity(frequency)**: Mild (<25 lesions): 108 (88MPX only + 20MPX/VZV), Moderate (26–100 lesions): 439 (353MPX only + 86MPX/VZV), Severe (101–250 lesions): 180 (141MPX only + 39MPX/VZV), Grave (>250 lesions): 56 (49MPX ONLY + 7MPX/VZV)	NA	NA	214 bacterial infection (176MPX only + 38MPX/VZV), 55 keloids (49MPX only + 6MPX/VZV), 9 alopecia (6MPX only + 3MPX/VZV), 32 ocular opacity (28MPX only + 4MPX/VZV), 17 unilateral eye complication (16MPX only + 1MPX/VZV), 16 bilateral eye complication (11MPX only + 5MPX/VZ)	NA	785 (633MPX only + 152 MPX/VZV) Unknown
Mbala et al. [48]	4 Unknown	4 Pregnancy, 1 Malaria	Skin rash on head, neck, upper limb, lower limb, palms, soles **Rash severity(frequency):** Mild (<25 lesions): 1, Moderate (26–100 lesions): 2, Severe (101–250 lesions): 1	NA	NA	2 miscarriage, 1 fetal death	4 inpatient	4 Unknown
Osadebe et al. [17]	333 Unknown	NA	329/329 fever, 262/328 chills, 278/328 fatigue/malaise, 227/325 LAP, 316/332 skin rash, 243/322 headache, 246/325 sore throat, 170/321 pruritus, 75/328 nausea/vomiting, 79/328 conjunctivitis, 105/323 photophobia, 192/331 cough, 190/326 oral ulcer, 327/330 febrile prodrome, 95/327 bedridden **Rash location:** 330/333 face, 328/333 thorax, 326/332 upper limb, 191/197 lower limb, 87/333 genitalia involvement, 324/333 palms, 309/333 soles, 190/326 mouth involvement **LAP location:** 206/326 cervical, 191/327 axillary, 168/326 inguinal	NA	NA	NA	NA	333 Unknown
Nakoune et al. [49]	3 Unknown	NA	Fever, rash, facial edema One dead case: fever, flat confluent vesicular rash involving palms and soles, cervical LAP, bilateral conjunctivitis, facial edema, mucus membrane involvement complicated with pulmonary edema, and profound hypothermia **Rash location:** 2/3 upper and lower limb, 1/3 palms and soles, 2/3 mouth involvement **LAP location:** 2/3 cervical, 1/3 inguinal	NA	NA	NA	3 Inpatient	2 Recovered, 1 Death
Yinka-Ogunleye et al. [51]	42 Unknown	1 Immunosuppressive illness	Data of 1 case available: fever, headache, malaise, sore throat, generalized well-circumscribed papulopustular rashes on the trunk, face, palms, and soles of the feet and subsequent umbilication, ulcerations, crusting, and scab formation, oral and nasal mucosal ulcers, generalized LAP	NA	NA	NA	1 Inpatient	1 Death, 41 Unknown
Vaughan et al. [52]	2 Unknown	NA	Pustular skin rash, fever, LAP, 1/2 pruritus **Rash location:** 1/2 head and neck, 1/2 genitalia involvement, 1/2 palms, 1/2 mouth involvement	NA	NA	NA	NA	1 Recovered, 1 Unknown
Doshi et al. [18]	8 Yes, 215 No	40 VZV (immunologic response)	**Rash severity(frequency):** Mild (<25): 104, Moderate (26–100): 433, Severe (101–250): 180, Grave (>250): 55	NA	NA	NA	NA	223 Unknown
Raynolds et al. [53]	2 Unknown	NA	Case1: fever, pustular and umbilicated lesion all over the body, oral mucosa, palms, and genital area, sweats, chills, vomiting, cough, pruritis Case2: fever, myalgia, enlarged cervical lymph nodes, dysphagia, macular rash	NA	NA	NA	1 Inpatient	1 Recovered, 1 Unknown
Doshi et al. [19]	7 Unknown	NA	2 Fever, 3 rash	NA	35 days (one case available)	NA	1 Inpatient	5 Recovered, 2 Death
Ye et al. [54]	1 Unknown	NA	Fever, body pain, malaise, dysphagia, enlarged cervical lymph nodes, generalized vesicular skin eruptions	10	NA	NA	NA	1 Unknown
Yinka-Ogunleye et al.* [20]	118 Unknown	4 HIV	118/118 vesiculopustular rash, 81/92 fever, 42/67 myalgia, 61/77 headache, 45/77 sore throat, 57/78 pruritus **Rash location:** 68/71 face, 56/70 trunk, 56/70 upper limb, 63/69 lower limb, 44/65 genitalia involvement, 48/70 palms, 42/66 soles	Mean: 13	NA	NA	NA	111/122 Recovered, 7/122 Death (4 HIV, 2 secondary bacterial infection, 1 infantile death), 4/122 Unknown
Besombes et al. [55]	6 No	NA	Only rash (n = 4), fever, and maculopapular rash on the palms and soles (n = 2)	14 days (one case available)	NA	NA	NA	6 Unknown
Ogoina et al. * [21]	18 Unknown	NA	21 vesiculopustular rash, 19 fever, 13 chills, 13 malaise, 5 myalgia, 13 LAP, 12 headache, 9 sore throat, 14 pruritus, 3 nausea, 1 diarrhea, 4 conjunctivitis, 3 photophobia, 4 cough, 10 genital ulcer, 11 oral ulcer, 2 hepatomegaly, 5 pain, 2 dehydration, 2 vulva swelling, 2 tongue sore, 1 scrotal swelling, 2 poor appetite, 1 generalized LAP **Rash location** (data of 1 case available): head and neck, lower limb, mouth involvement	NA	NA	NA	8 Outpatient, 13 Inpatient	20/21 Recovered, 1/21 Death (suicide)
Hughes et al. [22]	534 Unknown (400MPX only + 134MPX/VZV)	134/534	290 LAP (232MPX only + 58MPX/VZV), 381 sore throat (304MPX only + 77MPX/VZV), 296 cough (234MPX only + 62MPX/VZV), 140 bedridden (116MPX only + 24MPX/VZV), 291 mouth involvement (235MPX only + 56MPX/VZV), Coinfected cases: fatigue (n = 15), chills (n = 107), headache (n = 99), myalgia (n = 90)	NA	NA	NA	NA	534 Unknown (400MPX only + 134MPX/VZV)
Ogoina et al. [56]	40 Unknown	40 HIV	36 fever, 25 malaise, 35 LAP, 40 skin rash, 19 headache, 18 sore throat, 15 pruritus, 3 nausea/vomiting, 9 conjunctivitis, 9 photophobia, 25 genital ulcer, 3 hepatomegaly, 2 scrotal edema First symptom: rash (n = 23), fever (n = 12), genital rash (n = 2) Rash size: smaller than 2 cm (n = 20), larger than 2 cm (n = 20) **Rash severity(frequency):** Lower than 100 (n = 16), 100 to 1000 rashes (n = 17), More than 1000 rashes (n = 7) Rash distribution: confluent (n = 4), semiconfluent (n = 15), discrete (n = 21) Rash characteristics: monomorphic (n = 25), pleomorphic (n = 15) **Rash location:** 39 face, 25 scalp, 10 eye, 37 trunk, 35 upper limb, 34 lower limb, 27 genitalia involvement, 22 palms, 20 soles, 15 oral involvement **LAP location:** 11 cervical, 10 axillary, 12 inguinal, 12 generalized, 5 submental	NA	Longer than 28 days (n = 7/31), shorter than 28 days (n = 24/31)	19 bacterial skin infection, 5 gastroenteritis, 4 sepsis, 3 bronchopneumonia, 3 encephalitis, 3 keratitis, 1 premature rupture of membrane at 16 weeks gestation, 1 intrauterine fetal death, 11 anxiety, and depression, 12/18 hyperpigmented atrophic scars, 7/18 atrophic scars, 6/18 patchy alopecia, 7/18 hypertrophic skin scars, 1 deformity of facial muscle	40 inpatient	35 Recovered, 5 Death (1 suicide, 2 bronchopneumonia, 2 HIV)
Whitehouse et al. [23]	97 Yes, 960 No	NA	852/1027 chills, 888/1029 malaise, 754/1003 myalgia, 876/1034 LAP, 793/1011 headache, 600/1012 pruritus, 250/1010 nausea, 210/1016 conjunctivitis, 332/999 photophobia, 561/1024 cough, 736/1032 dysphagia, 278/1021 bedridden **Rash location:** 1036/1057 head and neck, 1028/1057 trunk, 1026/1057 upper limb, 786/1057 lower limb, 300/1057 genitalia involvement, 1009/1057 palms, 885/1057 soles, 570/1018 buccal ulcer Median lesion count: 102, IQR (61–177) **Rash severity(frequency):** Mild (<25 lesions): 180, Moderate (26–100 lesions): 462, Severe (101–249 lesions): 392, Grave (>250 lesions): 23	NA	NA	NA	NA	8 Death, 1049 Unknown
Hobson et al. [57]	3 Unknown	NA	Fever, skin rash	6–16 days	Mean: 17.6	NA	3 Outpatient	3 Recovered
Ng et al. [58]	1 Unknown	NA	Nodular skin lesions, fever, chills, myalgia	NA	NA	NA	1 Inpatient	1 Unknown
Besombes et al. [24]	3 Yes, 96 No	1 HIV, 2 VZV, 8 Malaria	74/99 fever, 60/99 chills, 65/99 asthenia, 45/99 myalgia, 71/99 adenopathy, 99/99 skin rash, 66/99 headache, 63/99 sore throat, 66/99 pruritus, 58/99 vomitting, 64/99 conjunctivitis, 59/99 photophobia, 63/99 cough, 63/99 dyspnea, 47/99 genital ulcer, 61/99 oral ulcer, 64/99 bedridden	7 days	NA	3/18 septicemia, 4/18 bronchopneumonia, 6/18 dehydration,2/18 corneal ulceration, 3/18 cutaneous bacterial superinfection, 1/18 fistulation of axillary adenopathy, 4/18 keloid healing	NA	NA
Pittman et al. [59]	2 Yes	1 HIV	1/216 fever, 99/216 chills, 188/216 fatigue/malaise, 15/216 myalgia, 213/216 LAP, 215/216 skin rash, 52/216 headache, 171/216 sore throat, 13/216 nausea/vomiting, 13/216 diarrhea, 20/216 conjunctivitis, 108/216 cough, 16/216 dyspnea, 61/216 abdominal pain, 54/216 dysphagia, 21/216 joint pain, 53/216 oral ulcer, 17/ 216 hepatomegaly or splenomegaly or both, 43/216 sweats, 15/216 Ear pain,1/ 216 hard of hearing, 74/216 nasal congestion, 5/216 visual change, 15/216 chest pain, 1/216 swelling, 110/216 anorexia, 6/216 bleeding, 2/216 petechiae,25/216 back pain, 9/216 neck stiffness, 3/216 dizziness ****LAP location:**** 185/216 cervical, 32/216 axillary, 167/216 inguinal, 1/216 supraclavicular, 82/216 submandibular	NA	between 7 and 21 days	NA	NA	3 Deaths
Adler et al. [26]	1 Yes, 6 No	NA	7/7 pleiomorphic skin lesions (including papules, vesicles, pustules, umbilicated pustules, ulcerating lesions, and scab), 3/7 fever, 1/7 night sweats, 1/7 groin swelling, 1/7 coryzal illness, 1/7 headache, 5/7 LAP **Rash location:** 7/7 head and neck, 7/7 trunk, 7/7 Limbs, 5/7 genitalia involvement, 4/7 palms, 2/7 soles **Rash severity:** 1/7 (10 lesions), 5/7 (1/7 30 lesions, 1/7 32 lesions, 1/7 40 lesions, 2/7 100 lesions), 1/7 (150 lesions)	NA	NA	3/7 low mood, 2/7 ulcerated inguinal lesion with delayed healing, 1/7 deep tissue abscesses, 1/7 severe pain, 1/7 conjunctivitis, 1/7 painful disruption of thumbnail from subungual lesion, 1/7 pruritis, 1/7 contact dermatitis from cleaning products	7/7 Inpatient	7/7 Recovered
Tarín-Vicente et al. [27]	32 Yes, 149 No	72/181 HIV, 31/181 any sexually transmitted infection (1/181 HIV, 10/181 chlamydia, 6/181 gonorrhoea, 2/181 herpes simplex virus, 2/181 Mycoplasma genitalium, 13/181 syphilis)	131 fever, 147 malaise/fatigue, 153 LAP, 181 skin rash, 96 headache, 66 sorethroat, 100 genitalia lesion, 45 oral ulcer, 147 Influenza-like illness, 160 at least one systemic feature, 87 systemic symptoms before the rash onset **Rash location:** 104/181 trunk, 104/181 limbs, 100/181 genitalia involvement, 108/181 palms and soles, 45/181 oral involvement, 51/181 perioral involvement ****Rash severity:**** 145/181 (mild 3–20 lesions), 21/181 (minimal 1–2 lesions), 15/181 (moderate > 20) ****LAP location:**** 53/181 cervical, 2/181 axillary, 110/181 inguinal	7 days	10 days	NA	178/181 Outpatient, 3/181 Inpatient	181/181 Recovered
Vallejo-Plaza et al. [60]	NA	2688/7393 HIV	5362/7040 general sign and symptoms (asthenia, fever, headache, muscle pain or odynophagia.), LAP (localized: 3504/7040 (women 72/153, men 3432/6887), generalized: 374/7040 (women 6/153, 368/6887)), overall 4711/7040; (78/153 women, 4633/6887men) genital ulcer, overall: 1333/7040 (women: 26/153, men 1307/6887) oral ulcer	NA	NA	secondary bacterial infections, oral ulcers, keratitis	244/7393 Inpatient	0 Death
Thornhill et al. [61]	49 Yes, 497 No	218/528 HIV, 109/377 STI; 32/377 gonorrhea, 20/377 chlamydia, 33/377 syphilis, 3/377 herpes simplex virus infection, 2/377 lymphogranuloma venereum, 5/377 chlamydia and gonorrhea, 14/377other or not stated 6/528 hepatitis B virus surface antigen positive, 30/528 hepatitis C virus antibody positive 8/528 hepatitis C RNA positive	330/528 fever, 216/528 fatigue/malaise, 165/528 myalgia, 295/528 LAP, 500/528 skin rash, 145/528 headache, 3/528 patients with conjunctival mucosa lesion, 164/528 genital ulcer, 66/528 oral ulcer, 113/528 Pharyngitis, 54/528 low mood, 75/528 proctitis or anorectal pain **Rash location:** 134/528 head and neck, 292/528 trunk or limbs, 383/528 anogenital area, 51/528 palms or soles, 383/528 anogenital area **Rash severity:** 207/528 (<5 lesions), 131/528 (5–10 lesions), 112/528 (11–20 lesions), 56/528 (>20 lesions), 22/528 (no lesion or missing data)	7 days	NA	1/528 epiglottitis, 2/528 myocarditis	458/528 Outpatient, 70/528 Inpatient	0 Death
Perez-Duque et al. [62]	NA	14/26 HIV	13 fever, 7 fatigue, 14 exanthoma (rash), 7 headache, 6 genital anal/ulcer, 5 anal ulcer/vesicle, 4 cervical LAP, 2 axillary LAP, 14 inguinal LAP	NA	NA	NA	NA	27/27 Recovered
Angelo et al. [28]	16 Yes, 166 No, 44 Unknown	92/209 HIV, 9/193 gonorrhoea, 5/193 primary or secondary syphilis, 4/193 chlamydia, 4/193 herpes simplex virus infection, 3/193 latent syphilis, 1/193 lymphogranuloma venereum, 1/193 molluscum contagiosum, 1/193 mycoplasma genitalium, 1/193 other syphilis, 1/193 streptococcal urethritis	131/226 fever, 50/226 chills, 93/226 fatigue or malaise, 32/226 myalgia, 134/219 LAP, 221/223 skin rash, 35/226 headache, 54/226 sore throat, 18/226 pruritus, 3/226 nausea, 13/226 diarrhea, 2/226 conjunctivitis, 16/226 cough, 1/226 abdominal pain, 124/226 genital ulcer, 43/221 oral ulcer **Rash location**: 51/224 face, 20/224 head, 7/224 neck, 57/215 trunk, 56/213 upper limb, 101/221 genitalia involvement, 25/224 palms, 9/225 soles, 43/221 mouth involvement, 60/218 anal region **LAP location:** 45/131 cervical, 5/131 axillary, 92/131 inguinal, 13/131 submandibular, 4/131 other	8 days	NA	NA	196 Outpatient, 30 Inpatient	226/226 Recovered
Betancort-Plata et al. [29]	NA	27/42 HIV, 11/24 other infectious diseases: (Chlamydia trachomatis, Haemophilus parainfluenza, Herpesvirus type 2MS Staphylococcus aureus, Mycoplasma genitalium, Mycoplasma hominis, Pantoea septica, Streptococcus dysgalactiae, Ureaplasma urealyticum)	15/42 fever, 17/42 LAP, 42/42 skin rash, 7/42 sore throat, 5/42 proctitis, 5/42 genital edema **Rash location:** 25/42 head and neck, 20/42 trunk, 27/42 upper limb, 16/42 lower limb, 22/42 genitalia involvement, 15/42 mouth involvement, 6/42 anal region **LAP location:** 7/42 cervical, 10/42 inguinal	NA	NA	NA	NA	NA
Rodríguez-Cuadrado et al. [63]	NA	NA	1/5 had lesions involving the conjunctiva, 1/5 oral ulcer	NA	NA	NA	NA	9/9 Recovered
Caria et al. [30]	3 Yes, 38 No	25/41 HIV, 5/41 gonorrhoea, 2/41 chlamydia, 1/41 syphilis, 3/41 HCV 3/41	21/41 fever, 9/41 fatigue/malaise, 9/41 myalgia, 19/41 LAP, 41/41 skin rash, 6/41 headache, 11/41 sore throat, 11/41 dysphagia, 25/41 genital ulcer, 2/41 proctitis **Rash location:** 13/41 (face and/or mouth), 23/41 (trunk and/or limbs), 25/41 (anogenital) **Rash severity:** 19/41 (<5 lesions), 10/41 (5–10 lesions), 9/41 (11–20 lesions), 3/41 (>20 lesions)	NA	NA	NA	37 Outpatient, 4 Inpatient	41/41 Recovered
Vanhamel et al. [31]	8 Yes, 131 Unknown	40/124 HIV, 2/139 STI	97/134 fever, 97/134 myalgia, 54/134 LAP, 134/134 skin rash, 97/134 headache, 3/134 sore throat, 3/134 cough **Rash location:** 102/134 anogenital	21 days	NA	NA	131 Outpatient, 8 Inpatient	NA
Fink et al. [32]	3 Yes, 153 No	47/155 HIV, 3/112 HBV, 2/116 HCV, 10/156 immunosuppressed, 20/156 gonorrhoea, 15/156 chlamydia, 5/156 chlamydia and gonorrhea, 16/156 herpes simplex virus, 17/156 syphilis	109/156 fever, 49/155 myalgia, 102/153 LAP, 2/156 sore throat, 3/10 diarrhea, 5/156 conjunctivitis, 1/10 dyspnea, 44/156 proctalgia, 8/156 genital edema, 16/156 upper respiratory tract disease affecting swallowing or airways, 8/156 severe genital pain **Rash severity:** 52/152 (0–10 lesions), 81/152 (11–100 lesions), 19/152 (>100 lesions)	NA	NA	NA	156 Inpatient	156/156 Recovered
Thornhill et al. * [64]	10 Yes, 129 No or not known	37/136 HIV, 17/131 STI; (7 syphilis, 6 chlamydia, 6 gonorrhoea)	76/122 presented with systemic features, 124/ 134 had skin lesions at presentation (105/121 were vesiculopustular, 95/129 at least one anogenital lesion, 65/119 had mucosal lesions, 17/129 perioral lesions, 19/130 Oral mucosal involvement), 84/132 fever, 65/127 fatigue/malaise, 38/128 myalgia, 68/132 LAP, 124/134 skin rash, 37/130 headache, 40/133 sore throat, 15/127 joint pain, 95/129 anogenital lesion, 62/131 perianal skin lesion, 33/115 anorectal, 31/129 oral ulcer, “1/134 meningism, encephalitis, or seizure, 30/117 low mood or anxiety **Rash location:** 31/129 oral lesion, 17/129 perioral skin lesion, 39/131 face, 19/131 lips or oral, 1/131 ocular, 60/130 trunk, 95/129 anogenital lesion, 41/131 vulvar skin lesion, 24/126 vaginal, 1/120 urethral, 36/129 palms or soles, 14/127 pharyngeal, 62/131 perianal skin lesion, 33/115 anorectal **Rash severity**: 126/136 median of 10 lesions	7 days	NA	cellulitis, abscess, bacterial superinfection; altered mental status and worsening left-sided weakness	17 Inpatient	0 Death
Patel et al. [25]	NA	70/197 HIV, 34/161Neisseria gonorrhoeae,18/161 Chlamydia trachomatis, 11/157 herpes simplex virus, 6/163 Treponema pallidum, 56/178 any STI	122/197 fever, 46/197 fatigue/malaise, 62/197 myalgia, 114/197 LAP, 27/197 skin rash, 49/197 headache, 33/197 sore throat, 27/197 pruritus, 2/197 conjunctivitis, 21/197 joint pain, 111/197 genital ulcer, 27/197 oral ulcer, 21/197 back pain, 71/197 rectal pain or pain on defecation, 31/197 penile swelling, 22/197 bleeding per rectum **Rash location:** 71/197 head and neck, 70/197 trunk, 74/197 arms/legs, 56/197 hands/feet, 11/197 genitalia involvement, 27/197 oropharyngeal, 82/197 anus or perianal area **Rash severity:** 22/168 (1 lesion), 102/168 (2–10 lesions), 36/168 (11–50 lesions), 0/168 (51–100 lesions), 8/168 (>100 lesions) **LAP location:** 16/197 cervical, 1/197 axillary, 90/197 inguinal, 7/197 cervical and inguinal	NA	NA	3/197 perianal or groin abscesses,2/197 tonsillar abscesses,1/197 Urinary retention 1/197 superimposed bacterial lower respiratory tract infection 1/197 disseminated lesions in the context of immunocompromise 5/197 substantial proctitis 1/197 rectal perforation 1/197 necrotizing secondary bacterial infection	25 Inpatient	0 Death
Girometti et al. [33]	NA	13/54 HIV	31/54 fever, 36/54 fatigue/malaise, 16/54 myalgia, 30/54 LAP, 54/54 skin rash, 11/54 sore throat, 5/5 pruritus, 33/54 genital ulcer, 4/54 oral/perioral ulcer **Rash location:** 11/54 head and neck, 14/54 trunk, 11/54 limbs, 33/54 genitalia involvement, 11/54 palms, 4/54 mouth involvement, 24/54 anal region **LAP location:** 2/54 cervical, 30/54 inguinal	NA	NA	NA	49 Outpatient, 5 Inpatient	54/54 Recovered
Vivancos-Gallego et al. [65]	5 Yes, 9 No, 11 Unknown	25/25 HIV	14/25 fever, 12/25 fatigue, 4/25 myalgia, 21/25 LAP, 25/25 skin rash, 5/25 sore throat, 13/25 proctitis **Rash location:** 10/25 head and neck, 12/25 trunk, 10/25 upper limb, 14/25 genitalia involvement, 5/25 palms, 3/25 mouth involvement	NA	8 days	NA	25 Outpatient	25/25 Recovered
Sheffer et al. [66]	NA	25/203 HIV, 11/203 STI (6/203 chlamydia, 2/203 syphilis,3/203 gonorrhea)6/203 inflammatory bowel disease	140/203 flu-like symptoms, 125/203 fever, 115/203 LAP, 186/203 skin rash, 14/203 sore throat, 33/203 anal/rectal pain **Rash location:** 48/203 face, including lips and mouth, 35/203 trunk, 44/203 arms/ hands, 128/203 genital/anal/perianal	NA	19.5 days	NA	NA	NA
Antinori et al. [67]	NA	2/4 HIV	2 fever, 1 myalgia, 4 rash (asynchronous), lesions appeared 1–3 days after systemic symptom **Rash location:** 1/4 head and neck, 3/4 trunk, 1/4 upper limb, 3/4 lower limb, 3/4 genitalia involvement, 1/4 soles, 2/4 anal region	NA	NA	NA	4 Inpatient	NA
Orviz et al. [34]	12 Yes, 36 No	19/48 HIV, 6/48 gonorrhea, 4/48 syphilis, 1/48 Mycoplasma genitalium (proctitis)	45/48 vesicular-umbilicated skin lesions, 25/48 fever, 32/48 asthenia, 25/48 myaligia, 39/48 LAP, 25/48 headache, 13/48 proctitis, 7/48 urethritis, 4/48 rash, 4/48 nasal congestion, 8/48 cough **Rash location:** 12/48 face, 16/48 trunk, 20/48 upper limb, 10/48 lower limb, 26/48 genitalia involvement, 2/48 palms and soles, 9/48 mouth involvement, 17/48 anal region **LAP location:** 4/48 cervical, 30/48 inguinal, 4/48 submandibular, 1/48 retroauricular	NA	NA	None	47 Outpatient, 1 Inpatient	NA
Hoffmann et al. [35]	28 Yes, 196 No	141/301 HIV, 234/265 HBV, 3/301 psoriatic arthritis, 2/301 DM, 2/301 HTN, 2/301 asthma, 177/301 previous STI	168/274 fever, 126/270 myalgia, 116/263 LAP, 276/279 skin rash, 126/270 headache, 53/266 night sweats **Rash location:** 72/296 head and neck, 122/292 trunk and limbs, 146/298 genitalia involvement, 152/299 anal region **Rash severity:** 216/276 (1–50 lesions), 60/276 (>50 lesions)	NA	NA	NA	286 Inpatient, 15 Outpatient	301/301 Recovered
Cobos et al. [36]	30 No	14/30 HIV, 4/30 syphilis, 1/30 HCV	23/30 fever, 18/30 fatigue, 12/30 myalgia, 23/30 LAP, 29/30 skin rash, 16/30 headache, 8/30 sore throat, 7/30 joint pain, 16/30 genital ulcer, 10/30 Oral/perioral ulcer, 9/30 proctalgia/proctitis **Rash location:** 24/30 trunk, 14/30 upper limb, 16/30 lower limb, 16/30 genitalia involvement, 5/30 palms, 6/30 soles, 10/30 mouth involvement, 10/30 anal region **LAP location:** 10/30 cervical, 3/30 axillary, 18/30 inguinal	7.5 days	3–4 weeks	NA	29 Outpatient, 1 Inpatient	30/30 Recovered
Hoffmann et al. [37]	49 Yes, 333 No, 164 Unknown	256/546 HIV, 7 psoriatic arthritis, 66/481 had a history of hepatitis C virus infection, 50/469 hepatitis B infection, STI diagnosed during the last 6 months (syphilis, chlamydia, 32.4% gonorrhea)	272/511 fever, 208/507 headache and pain in the limbs, 73/503 night sweats, 213/500 lymph node swelling **Rash location:** 123/523 oral, perioral, head and neck, 196/522 trunk and/or extremities, 267/535 genitalia involvement, 257/536 anal region **Rash severity:** 223/493 (1–3 lesions), 181/493 (4–10 lesions), 74/493 (11–50 lesions), 11/493 (>50 lesions)	NA	NA	22/546 severe clinical complications (massive swelling of lymph nodes and genitals, extensive involvement of the entire integument, haemorrhage, refractory pain)	520 Outpatient, 26 Inpatient	0 Death
Van Ewijk et al. [38]	126 Yes, 822 No, 52 Unknown	187/882 HIV, 56/882 STI	521/991 fever, 268/991 fatigue, 257/991 myalgia, 914/991 skin rash, 322/991 headache, 134/991 pruritus, 45/991 diarrhea, 43/991 cough, 179/991 proctalgia, 70/991 back pain, 81/991 respiratory symptoms **Type of lesions:** 31% maculopapular, 59% vesicular, 46% pustular, 12% crusts, 5% other types **Rash location:** 312/991 head and neck, 350/991 trunk, 469/991 limbs, 469/991 genitalia involvement, 105/991 mouth involvement, 305/991 anal region	12 days	NA	NA	941 Outpatient, 11 Inpatient	NA
Mailhe et al. [39]	29 Yes, 209 No, 26 Unknown	63/256 HIV	171/253 fever, 174/251 adenopathy, 51/252 pharyngitis, 41/252 angina, 31/255 respiratory signs, 89/255 headaches, skin lesion (82/244 papules, 138/243 vesicles, 80/243 pustular papules, 84/244 ulcerations, 59/243 scabs, 20/248 rash) **Rash location:** 88/252 face, 105/252 trunk, 121/252 limbs, 135/252 genitalia involvement, 36/250 palmoplantar areas, 100/251 perianal	6 days	NA	45/257 anal pain, 25/257 secondary bacterial skin infections such as cellulitis, 89/255 headaches, 11/257 urinary signs, 10/257 ocular disease, 7/257 abscess, 5/257 lymphangitis, 3/257 paronychia, 1 Bell’s palsy	17 Inpatient	0 Death
Núñez et al. [40]	NA	299/565 HIV, 27/565 syphilis, 2/565 gonorrhea, 1/565 chronic hepatitis B, 10/565 chronic hepatitis C, 4/565 unspecified chronic hepatitis	564/565 rash, 446/565 fever, 64/565 shivers, 232/565 lymphadenopathies, 191/565 fatigue, 248/565 headache, 10/565 nausea, 6/565 vomitting, 270/565 myalgias, 241/565 articular pain, 71/565 lower back pain, 35/565 proctitis, 5/565 conjunctivitis, 1/565 photophobia, 40/565 cough, 140/565 odynophagia, skin lesions morphology (213/565 macules, 332/565 papules, 297/565 vesicles, 250/565 pustules, 137/565 scabs, 58/565 ulcers) **Rash location:** 74/530 scalp, 192/530 face, 62/530 neck, 249/530 chest, 28/530 back, 81/530 abdomen, 279/530 upper limb, 209/530 lower limb, 280/530 Genital and/or perianal area, 24/530 palms, 21/530 soles, 42/530 mouth involvement **LAP location**: 98/538 cervical, 14/538 axillary, 127/538 inguinal, 8/538 Retroauricular, 11/538 submandibular, 1/538 occipital	8 days	NA	NA	6 Inpatient	1 Death
Maldonado-Barrueco et al. [68]	NA	19/30 HIV, 22/30 syphilis, 3/30 HBV, 1/30 HCV, 4/30 Treponema pallidum, 8/30 herps simplex virus-I, 4/30 herpes simplex virus-II, 3/30 CT- Lymphogranuloma venereum (LGV), 9/30 Neisseria gonorrhoeae (NG), 8/30 Chlamydia trachomatis (CT), 3/30 Mycoplasma genitalium (MG)	15/30 LAP, 9/30 proctitis, 4/30 tenesmus, 21/30 fever, 5/30 sore throat, 10/30 discharge, 10/30 pruritus, 30/30 lesion, 18/30 pain of lesions, 9/30 pruritic mpox lesions, 1/30 acute epiglottitis with air compromise **Rash location:** 4/30 trunk, 6/30 penis/pubis, 2/30 nose/lip/mouth, 11/30 perianal	NA	NA	NA	NA	NA
Catala et al. [41]	20 Yes, 145 No, 20 Unknown	78/185 HIV	100/185 fever, 81/185 fatigue/malaise, 104/185 LAP, 59/181 headache, 34/185 sore throat, 0/185 nausea/vomit, 0/185 abdominal pain, 21/185 joint pain, 128/185 genital ulcer, 10/185 oral/perioral ulcer, 40/185 proctalgia/proctitis, 12/185 back/lumbar pain, type of lesions: 12/185 macular, 90/185 papular, 54/185 vesicular, 138/185 pustular or pseudopustular **Rash location**: 74/185 head and neck, 104/185 trunk, 70/185 upper limb, 52/185 lower limb, 128/185 genitalia involvement, 12/185 palms, 22/185 soles, 26/185 mouth involvement, 62/185 anal region **Rash severity:** 21/185 (<2 lesions), 152/185 (2–25 lesions), 11/185 (26–100 lesions), 1/185 (>100 lesions)	6 days	NA	NA	181 Outpatient, 4 Inpatient	185/185 Recovered
Relhan et al. [69]	5 No	1/5 positive for hepatitis B virus surface antigen (HBsAg), 0/5 hepatitis-C, 0/5 herpes simplex virus 1&2, 0/5 syphilis	5/5 fever, 1/5 chills, 0/5 fatigue, 5/5 myalgia, 4/5 LAP, 5/5 skin rash, 1/5 headache, 2/5 sore throat, 2/5 dysphagia, 0/5 joint pain, 3/5 oral ulcer, 2/5 dysuria, 2/5 genital swelling, 1/5 chest pain **Rash location:** 5/5 head and neck, 5/5 trunk, 5/5 upper limb, 4/5 lower limb, 4/5 genitalia involvement, 3/5 palms, 2/5 soles, 3/5 mouth involvement **Rash severity:** 5/5 (20–100 lesions) **LAP location:** 1/5 cervical, 4/5 inguinal, 1/5 generalized, 1/5 submental, 1/5 retro-auricular,1/5 submandibular	NA	NA	NA	5 Inpatient	5/5 Recovered
Cassir et al. [42]	15 Yes, 121 No	14/30 HIV, Chlamydia trachomatis 4/127, Neisseria gonorrhoeae 13/127, Mycoplasma genitalium 1/127, Syphilis 4/127, Trichomonas vaginalis 1/127, other STIs 19/127	72/136 fever, 51/136 LAP, 115/136 skin rash, 11/136 sore throat, 68/136 genital ulcer, 3/136 oral/perioral ulcer, 30/136 proctalgia/proctitis, 5/136 genital edema, 44/136 influenza-like illness, type of lesion: 26/136 papular, 60/136 vesicular, 25/136 pustular, 10/136 scabbed **Rash location**: 28/136 trunk, 13/136 limbs, 68/136 genitalia involvement, 25/136 mouth involvement, 63/136 anal region **Rash severity**: 98/136 (1–10 lesions), 17/136 (11–50 lesions), 0/136 (>50 lesions) **LAP location:** 26/136 cervical, 28/136 inguinal, 4/136 generalized	NA	NA	Proctitis 30/136, tonsillitis 5/136, penile edema 5/136, bacterial skin abscess 4/136, exanthem 3/136	130 Outpatient, 6 Inpatient	136/136 Recovered
Pascom et al. * [70]	NA	2825/6000 HIV, 412/3728 syphilis, 214/3728 genital herpes, 41/3728 chlamydia, 38/3728 gonorrhea, 37/3728 genital warts, 115/3728 other	4709/8167 fever, 1070/8167 chills, 2679/8167 fatigue/malaise, 2919/8167 myalgia, 2919/8167 LAP, 3498/8167 skin rash, 3280/8167 headache, 1108/8167 sore throat, 470/8167 nausea/vomiting, 84/8167 diarrhea, 81/8167 conjunctivitis, 167/8167 photophobia, 300/8167 cough, 398/8167 joint pain, 1542/8167 genital ulcer, 359/8167 oral/perioral ulcer, 338/8167 proctitis, 1586/8167 back/lumbar pain, 332/8167 genital edema, 3255/8167 adenomegaly, 265/8167 mucosal lesion, 99/8167 hemorrhage **Rash location:** 2260/8176 head and neck, 2893/8167 trunk, 2832/8167 upper limb, 2092/8167 lower limb, 3913/8167 genitalia involvement, 867/8167 palms, 404/8167 soles, 806/8167 mouth involvement, 1585/8167 anal region	NA	NA	NA	7786 Outpatient, 365 Inpatient, 16 ICU	3109/8167 Recovered, 3/8167 Death, 5055/8167 Unknown
Martins-Filho et al. [71]	NA	NA	6944/7518 cutaneous lesions, 4622/7518 genital and anal lesions, 4353/7518 fever, 3088/7518 LAP, 2980/7518 headache, 2773/7518 myalgia, 2511/7518 asthenia, 807/7518 sore throat, 662/7518 oral lesions, 227/7518 arthralgia, 158/7518 proctitis, 73/7518 photosensitivity, 36/7518 conjunctivitis	NA	NA	NA	NA	NA
Wong et al. [72]	7 Unknown	7/7 HIV	1/7 fever, 6/7 fatigue, 3/7 LAP, 7/7 skin rash **Rash location:** 5/7 head and neck, 6/7 upper limb, 6/7 lower limb, 6/7 genitalia involvement, 6/7 palms and soles, 1/7 anal region **Rash severity:** 7/7 (<25 lesions)	NA	2 weeks	NA	5 Outpatient, 2 Inpatient	7/7 Recovered
Suner et al. [43]	2 Yes, 73 No	39/77 HIV, 25/77 gonorrhea, 13/77 chlamydia, 1/77 Lymphogranuloma venerum, 20/77 syphlis, 1/77 herpes simplex virus, 3/77 warts	49 fever, 55 fatigue, 64 lymphadenopathies, 69 skin rash, 51 headaches, 33 sorethroat, 3 nausea, 2 abdominal pain, 43 joint pain, 24 proctitis; 17 tonsilitis **Rash location:** 31/77 head and neck, 46/77 trunk, 47/77 upper limb, 36/77 lower limb, 36/77 genitalia involvement, 18/77 mouth involvement, 21/77 anal region	6 days	25 days	29/64 total, 13/77 scars, 17/77 bacterial skin abscess, 7/77 penile edema	76 Outpatient, 1 Inpatient	NA
Cash-Goldwasser et al. [73]	5 Unknown	2/5 HIV	Ocular manifestation (4 conjunctivitides, 2 photophobia, 5 redness, 5 eye pain, 4 unilateral eye symptoms, 1 bilateral eye symptom, including pain, itching, swelling, discharge, foreign body sensation, photosensitivity, and vision changes, multiple right eyelid lesions, periorbital swelling) **Rash location:** 3/5 head and neck, 4/5 trunk, 2/5 upper limb, 1/5 lower limb, 2/5 genitalia involvement, 2/5 anal region	NA	10 days	NA	1 Inpatient	5/5 Recovered
Ciccarese et al. [74]	1 Yes, 15 No	3/16 HIV	4/16 fever, 3/16 LAP, 14/16 skin rash, 1/16 sore throat, 3/16 pruritus, 2/16 genital ulcer, 1/16 genital pain, 1/16 anorectal pain, 1/16 perianal pain, 2/16 anal pain, 1/16 nasal pain, 1/16 anal itch **Rash location:** 1/16 head and neck, 1/16 trunk, 1/16 upper limb, 1/16 lower limb, 2/16 genital involvement, 4/16 anal region **Rash severity:** 8/16 (<10 lesions), 4/16 (10–20 lesions), 2/16 (>20 lesions) **LAP location:** 1/16 cervical, 3/16 inguinal	NA	NA	NA	12 Outpatient, 4 Inpatient	16/16 Recovered
Aguilera-Alonso et al. [75]	NA	NA	16/16 skin lesion, 5/16 LAP, 4/16 fever, 2/16 fatigue, 2/16 sore throat, 1/16 myalgia, 1/16 vomiting, 1/16 diarrhea	NA	NA	1/16 bacterial superinfection that required drainage of an abscess	16 Outpatient	16/16 recovered
Gnanaprakasam et al. [76]	NA	10/23 HIV, 1/23 chlamydia, 1/23 gonorrhea, 1/23 syphilis	10/23 fever, 10/23 genital ulcer, 4/23 oral ulcer, 2/23 Abscess, 2/23 urethritis, 4/23 proctitis **Rash location:** 10/23 genitalia involvement, 4/23 mouth involvement, 6/23 anal region	NA	NA	2/23 severe proctitis and fever	2 Inpatient	NA
Choudhury et al. [77]	32 Yes, 119 No, 28 Unknown	55/131 HIV	91/179 fever, 88/179 flu-like symptoms, 45/179 myalgia, 73/179 LAP, 159/179 skin rash, 59/179 headache, 82/179 pain	7 days	NA	NA	169 Outpatient, 9 Inpatient, 1 Unknown	NA
Srichawla et al. [78]	9 Unknown	1/9 HIV, 2/9 syphilis, 3/9 gonorrhea, 1/9 HSV, 1/9 chlamydia	8/9 presented with an acute-onset eruption of umbilicated lesions, 1/9 presented with severe subacute-onset migraine headaches with associated photophobia and phonophobia, 5/9 fever, 1/9 fatigue/malaise, 2/9 myalgia, 9/9 skin rash, 3/9 headache, 1/9 sore throat, 5/9 genital ulcer, 2/9 oral ulcer, 1/9 dizziness **Rash location:** 3/9 head and neck, 3/9 trunk, 2/9 upper limb, 2/9 lower limb, 5/9 genitalia involvement, 2/9 mouth involvement, 3/9 anal region	NA	NA	1/9 Cellulitis of the left upper extremity, 1/9 Cellulitis of the scrotum	7 Outpatient, 2 Inpatient	9/9 Recovered
Maldonado et al. [44]	NA	136/205 HIV, 67/205 self-reported history of syphilis, 10/205 other self-reported STIs (genital warts, hepatitis C, gonorrhea, and genital herpes)	111/205 experienced systemic symptoms before the skin lesions, 162/205 fever, 123/205 malaise, 119/205 headache, 105/205 fatigue, 79/205 sore throat, 7/205 cough, 6/205 rhinorrhea, 111/205 LAP, 174/205 skin lesion, 69/205 chills and sweats, 27/205 pruritus, lesion morphology (116/205 macular, 162/205 papular, 44/205 vesicle, 177/205 pustule, 4/205 ulcer, 65/205 crust), lesion pattern (31/205 monomorphic, 174/205 polymorphic) **Rash location:** 128/205 (face, head or neck), 142/205 trunk, 111/205 upper limb, 86/205 lower limb, 160/205 anogenital **LAP location:** 18/205 unilateral cervical, 22/205 bilateral cervical, 2/205 axillary, 34/205 unilateral inguinal, 29/205 bilateral inguinal, 13/205 generalized, 2/205 submandibular	7 days	NA	18/205 bacterial superinfection, 2/205 vasculitis of small vessels, 19/205 proctitis, 4/205 balanitis, 3/205 necrosis of skin lesion, 2/205 generalized exanthem, 1/205 orchiepididymitis	184 Outpatient, 21 Inpatient	NA
Rekik et al. [79]	1 Yes, 19 No	6/20 HIV, 4/20 herpes simplex virus	14/20 anal margin lesions, 16/20 anal canal lesions, 7/20 anal hypertonia, 12/20 rectal lesions, 18/20 fever, 19/20 fatigue/malaise, 14/20 myalgia, 13/20 LAP, 17/20 cutaneous lesions at sites other than the anus, 11/20 pruritus, 8/20 genital ulcer, 13/20 anal pain, 12/20 anal bleeding, 10/20 dyschezia, 13/20 tenesmus, 3/20 burning, 9/20 swelling, and 9/20 mucus discharge, 1/20 penile lymphangitis **Rash location:** 8/20 genitalia involvement, 16/20 anal canal lesions, 14/20 anal margin lesions **LAP location:** 13/20 inguinal	NA	NA	1/20 severe ulcerative proctitis	1 Inpatient	NA
Prasad et al. [80]	4 Yes	38/101 HIV, 38/101 STI, 17/101 gonorrhea, 14/101 syphilis, 7/101 chlamydia, 7/101 herpes simplex virus	65/101 fever, 21/101 chills, 39/101 fatigue/malaise, 18/101 myalgia, 52/101 LAP, 99/101 skin rash, 19/101 headache, 21/101 sore throat, 1/101 ocular/ ophthalmic symptoms, 5/101 cough, 10/101 joint pain, 15/101 oral ulcer, 16/101 rectal pain, 5/101 penile edema, 17/101 proctitis **Anatomical site of mucocutaneous lesions**: 11/101 oral cavity,3/101 tonsils, 1/101 soft palate **Edema location**: 2/101 peri-orbital, 8/101 face, 4/101 peri-rectal/peri-anal, 12/101 scrotal/penile, 4/101 extremities. Day 0: most common rash types are papules, vesicles or pustules. Days 1–5: papules, vesicles, pustules, or erosions/ ulcers. Days 6–10: the most common were postules followed by erosions/ulcers and crusts/scabs. **Rash severity:** 77/101 mild (≤25 lesions), 20/101 moderate (26–100 lesions), 1/101 severe (101+ lesions)	7 days	20 days	1/101 sepsis	21 Inpatient	NA
Assiri et al. [81]	NA	0 HIV	7/7 skin lesions, 6/7 fever, 5/7 adenopathy, 3/7 penile lesions, 1/7 finger paronychia, 1/7 oral vesicles, 1/7 myalgia, 2/7 back pain, 1/7 fatigue, 5/7 LAP **Rash location:** 1/7 head and neck, 1/7 trunk, 1/7 limbs, 4/7 genitalia involvement, 4/7 palms, 3/7 soles, 1/7 mouth involvement **LAP location:** 2/7 cervical, 1/7 axillary, 1/7 occipital, 1/7 submandibular	NA	10 days	NA	NA	NA

NA: not available; LAP: lymphadenopathy. * Data were not defined in confirmed patients and were reported in the total population; however, available data were included according to the majority number of confirmed patients among the total population.

**Table 6 biomedicines-11-00957-t006:** Pooled frequency of the confirmed mpox patient characteristics.

		Pre-2022 Outbreak						Post-2022 Outbreak					
	Variables	Number of Studies	% Pooled Frequency (95% CI)	n/N (%)	Publication Bias (*p*-Value)	Heterogeneity Test		Number of Studies	% Pooled Frequency (95% CI)	n/N (%)	Publication Bias (*p*-Value)	Heterogeneity Test	
						I^2^(%)	*p*-Value					I^2^(%)	*p*-Value
**Gender**	Male	12	59.1 (54.6–63.6)	1802/3131 (57.55)	0.73	77.35	0.00	19	98.7 (97.1–99.4)	9913/10,796 (91.82)	0.00	92.19	0.00
	Female	12	40.9 (36.4–45.4)	1329/3131 (42.44)	0.73	77.35	0.00	19	1.3 (0.7–2.5)	565/10,796 (5.23)	0.03	86.77	0.00
**MSM** (Among the total population)		0	-	-	-	-	-	19	93.5 (91.0–95.4)	8265/9151 (90.31)	0.29	86.60	0.00
**Smallpox vaccination status**	Yes	7	5.6 (2.8–10.6)	137/2253 (6.08)	0.76	86.92	0.00	14	11.0 (8.9–13.6)	343/2967 (11.56)	0.00	67.70	0.00
	No	7	94.4 (89.4–97.2)	2116/2253 (93.91)	0.76	86.92	0.00	14	88.4 (85.4–90.8)	2604/2967 (87.76)	0.01	74.95	0.00
**Clinical manifestations**	Fever	8	82.7 (49.4–95.9)	1138/1399 (81.34)	0.90	94.22	0.05	19	62.3 (58.0–66.4)	7253/12,276 (59.08)	0.83	91.20	0.00
	Chills	5	68.1 (49.6–82.2)	1286/1691 (76.04)	0.8	97.06	0.05	4	18.5 (11.6–28.3)	1253/9163 (13.67)	0.30	96.24	0.00
	Malaise/fatigue	6	72.1 (42.8–89.9)	1611/2326 (69.26)	0.45	99.13	0.13	13	51.8 (43.1–60.4)	3827/10,904 (35.09)	0.29	96.80	0.68
	Myalgia	5	39.2 (14.8–70.5)	861/1406 (61.23)	0.46	98.14	0.51	14	39.2 (33.6–45.2)	4144/11,450 (36.19)	0.91	94.28	0.00
	LAP	8	80.6 (68.7–88.6)	2214/2735 (80.95)	0.71	96.70	0.00	19	55.5 (49.7–61.1)	5262/12,242 (42.98)	0.10	95.05	0.06
	Skin rash	9	98.8 (96.2–99.6)	1633/1650 (98.96)	0.07	64.08	0.00	17	96.6 (90.8–98.8)	6899/11,914 (57.90)	0.02	98.53	0.00
	Headache	7	60.4 (40.7–77.1)	1228/1753 (70.05)	0.13	97.09	0.30	15	43.2 (38.1–48.4)	4777/11,878 (40.21)	0.84	93.37	0.01
	Sore throat/odynophagia	7	70.8 (65.3–75.8)	1305/1771 (73.68)	0.22	78.51	0.00	15	19.3 (14.3–25.6)	1618/10,451 (15.48)	0.13	94.93	0.00
	Pruritus	6	61.2 (54.1–67.8)	908/1538 (59.03)	1.00	72.07	0.00	4	12.8 (8.2–19.4)	184/1427 (12.89)	1.00	78.01	0.00
	Nausea/vomiting	6	21.2 (13.3–32.0)	499/2307 (21.62)	0.70	95.36	0.00	6	3.2 (2.0–5.1)	531/10,211 (5.20)	0.70	81.91	0.00
	Conjunctivitis	7	23.3 (15.3–33.7)	497/2320 (21.42)	1.00	94.21	0.00	6	1.6 (0.8–3.0)	103/9448 (1.09)	1.00	77.88	0.00
	Photophobia	4	37.0 (26.6–48.7)	499/1442 (34.60)	1.00	89.97	0.03	2	0.7 (0.1–7.5)	168/8732 (1.92)	-	83.42	0.00
	Cough	7	55.1 (51.3–58.8)	1499/2724 (55.02)	0.76	66.48	0.00	7	5.4 (3.7–7.6)	417/10,336 (4.03)	0.76	84.39	0.00
	Diarrhea	3	8.5 (4.5–15.2)	90/870 (10.34)	1.00	69.93	0.00	4	5.0 (1.6–14.4)	145/9394 (1.54)	0.73	97.08	0.00
	Abdominal pain	2	31.8 (28.8–35.1)	270/849 (31.80)	-	40.84	0.00	3	1.1 (0.4–3.2)	3/488 (0.61)	1.00	39.88	0.00
	Arthralgia	2	16.2 (6.0–37.1)	180/849 (21.20)	-	95.26	0.00	5	22.1 (5.6–57.6)	710/9024 (7.86)	1.00	99.52	0.11
	Proctalgia/proctitis	0	-	-	-	-	-	15	16.6 (10.3–25.6)	861/11,244 (7.65)	0.37	97.76	0.00
	Backache	1	-	25/216 (11.57)	-	-	-	6	12.0 (7.1–19.7)	1826/10,339 (17.66)	0.70	97.23	0.00
	Genital edema	1	-	3/21 (14.28)	-	-	-	6	6.8 (4.0–11.2)	372/8696 (4.27)	0.70	82.67	0.00
**Rash location**	Head & neck	4	98.0 (97.1–98.6)	1441/1468 (98.16)	0.73	3035	0.00	13	36.7 (29.7–44.3)	3542/11,594 (30.55)	0.50	96.85	0.00
	Trunk	4	95.2 (82.4–98.8)	1419/1467 (96.72)	0.73	93.32	0.00	16	45.2 (38.6–52.0)	4579/11,927 (38.39)	0.68	96.19	0.16
	Upper limb	4	94.9 (82.7–98.6)	1415/1466 (96.52)	0.73	92.92	0.00	16	41.8 (36.4–47.4)	4492/11,925 (37.66)	0.44	94.37	0.00
	Lower limb	4	91.0 (71.0–97.7)	1047/1330 (78.72)	1.00	92.77	0.00	15	36.2 (29.6–43.3)	3553/11,712 (30.33)	0.09	96.63	0.00
	Palms	4	85.9 (65.1–95.2)	1854/2100 (88.28)	0.80	97.77	0.00	10	15.4 (7.4–29.2)	1357/10,206 (13.29)	0.85	98.83	0.00
	Soles	5	73.8 (55.3–86.5)	1627/2096 (77.62)	0.80	97.48	0.01	8	10.6 (4.0–25.2)	608/9618 (6.32)	1.00	98.54	0.00
	Genitalia	6	53.5 (36.8–69.5)	488/1529 (31.91)	0.13	94.03	0.68	18	55.6 (51.7–59.4)	6027/12,127 (49.69)	0.15	88.11	0.00
	Oropharyngeal	7	52.4 (45.5–59.2)	1456/2713 (53.66)	0.36	90.31	0.49	13	18.3 (13.2–24.9)	1304/11,185 (11.65)	0.66	95.95	0.00
	Anal/perianal	0	-	-	-	-	-	16	39.8 (30.4–49.9)	2990/11,555 (25.87)	0.22	98.10	0.04
**Rash number (severity)**	Mild (<25)	5	8.5 (4.7–14.8)	244/2552 (9.56)	0.80	92.89	0.00	8	84.9 (71.9–92.5)	1411/1728 (81.65)	0.38	96.28	0.00
	Moderate (26–100)	5	53.6 (46.6–60.5)	1310/2552 (51.33)	0.80	88.72	0.31	7	15.2 (8.1–26.8)	272/1464 (18.57)	0.36	95.44	0.00
	Severe (101–249)	5	28.2 (20.5–37.4)	748/2552 (29.31)	1.00	93.70	0.00	3	1.6 (0.1–21.2)	20/518 (3.86)	1.00	88.16	0.00
	Grave (>250)	5	8.4 (5.5–12.6)	250/2446 (10.22)	0.80	85.30	0.00	1	-	0/181 (0)	-	-	-
**LAP location**	Cervical	3	79.7 (61.6–90.6)	934/1175 (79.48)	1.00	97.09	0.00	9	20.7 (15.8–26.8)	285/1365 (20.87)	0.34	78.88	0.00
	Axillary	4	45.7 (30.5–61.8)	819/1576 (51.96)	0.08	97.21	0.60	5	2.8 (1.4–5.4)	26/1085 (2.39)	0.80	58.08	0.00
	Inguinal	3	63.8 (50.7–75.1)	722/1175 (61.44)	1.00	94.30	0.04	9	44.1 (30.2–59.1)	508/1365 (37.21)	0.60	95.57	0.44
	Generalized	2	-	14/44 (31.81)	-	-	-	5	2.8 (0.8–9.6)	115/8832 (1.30)	0.80	95.69	0.00
**Coinfection**	HIV	3	11.4 (0.2–91.3)	5/227 (2.20)	0.29	88.75	0.36	19	41.1 (35.5–47.0)	4404/10,037 (43.87)	0.57	94.91	0.00
	VZV	2	-	286/1319 (21.86)	-	-	-	0	-	-	-	-	-
**Management status**	Outpatient	2	23.8 (4.0–69.9)	8/28 (28.57)	-	52.72	0.25	18	95.1 (92.9–96.6)	11581/12,273 (94.36)	1.00	87.49	0.00
	Inpatient	2	76.2 (30.1–96.0)	20/28 (71.42)	-	52.72	0.25	18	4.9 (3.3–7.1)	676/12,273 (5.50)	1.00	87.66	0.00
	ICU	2	3.7 (0.5–22.4)	0/28 (0.00)	-	0.00	0.00	9	0.2 (0.2–0.4)	16/9159 (0.17)	0.00	0.00	0.00
**Outcome**	Recovery	7	82.1 (46.9–96.0)	448/1525 (29.37)	0.22	90.77	0.07	12	99.4 (92.5–99.9)	5247/10,306 (50.91)	0.00	95.17	0.00
	Death	7	4.2 (1.6–11.0)	28/1525 (1.83)	0.54	83.22	0.00	12	0.2 (0.1–0.3)	4/10,306 (0.03)	0.00	19.18	0.00

LAP: lymphadenopathy; MSM: men having sex with men.

## Data Availability

All data were included in the manuscript.

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
