# Peer review of "Demographic, Epidemiologic, and Clinical Characteristics of Human Monkeypox Disease Pre- and Post-2022 Outbreaks: A Systematic Review and Meta-Analysis"

_biomedicines, 2023, doi:10.3390/biomedicines11030957_

Round 1

Reviewer 1 Report

Dear authors, i appreciated reading your manuscript, especially the Figure that is very relevant. However I have one major issue regarding the title and the aim of the present review of the litterature : it does conclude about the new outbreak of Mpox, meanwhile the review of the literature is prior the early outbreak which occured in May 22 ( A systematic search was performed for relevant studies published in Pubmed/Medline, Embase, and Scopus from 2012 up to May 2022)

Author Response

Dear reviewer,
Thank you for your attention and thoughtful comment. Our study aimed to conduct a systematic review and meta-analysis on the confirmed mpox patients, 10 years before the recent 2022 outbreak to provide a basis for ongoing investigations about the 2022 mpox outbreak and probable changes in demographic, epidemiologic, and clinical manifestations of the disease. In this regard, we tried to make some comparisons between our results and the ongoing growing evidence of 2022 cases in the Discussion part. However, now we revised and updated the manuscript and included relevant 2022 studies in the meta-analysis to make a more precise comparison between pre- and post-2022 mpox outbreaks.

Reviewer 2 Report

Thanks for the invitation to review the current manuscript.

In the current manuscript, Hossein Hatami et al. conducted a systematic review and meta-analysis to present an overview of the demographic, epidemiologic and clinical features of mpox cases in published literature over a period of 10 years.

This systematic review comes amid the ongoing 2022 multi-country mpox outbreak with more than 85000 cases recorded in more than 100 countries previously non-endemic in mpox worldwide.

The major limitation of the current review is that the search concluded in May 2022, therefore, the authors missed several important recent studies describing the demographic and clinical features of mpox. Thus, I recommend reconsidering the search strategy which would improve the final output of the review to a large extent. Additionally, this improved search strategy will be helpful to highlight the differences in mpox presentation over two periods (pre- and post-2022 mpox outbreak).

Minor comments:

-Please replace HMPX by the recommended nomenclature of the disease (mpox) as stated in:  https://www.who.int/news/item/28-11-2022-who-recommends-new-name-for-monkeypox-disease

-In the re-analysis, please try to consider the inclusion of grey literature and other sources (e.g., Google Scholar) as sources of search for the literature.

-In the Methods, please be more specific regarding the inclusion and exclusion criteria.

-Please inspect the manuscript carefully for minor typos, language and grammatical errors.

-Please try to improve the quality of Tables and Figures in terms of font size and style.

Author Response

Dear reviewer,

Thank you for your thoughtful comments. We have carefully edited the manuscript as you requested and provided a point-by-point response below.

We updated our review and extended it to the scientific articles that were published from May 1, 2022, to February 15, 2023. We included 44 more studies, 42 of which were according to the 2022 outbreak. Also, we performed a meta-analysis on the data obtained from both pre-and post-2022 outbreaks to highlight the differences in mpox characteristics over two periods.

-Please replace HMPX by the recommended nomenclature of the disease (mpox) as stated in:  https://www.who.int/news/item/28-11-2022-who-recommends-new-name-for-monkeypox-disease

Response: Dear reviewer, thank you for pointing out this issue. The manuscript was changed accordingly.

-In the re-analysis, please try to consider the inclusion of grey literature and other sources (e.g., Google Scholar) as sources of search for the literature.

Response: Thank you for your comment. We searched Google Scholar for the grey literature and included some relevant studies accordingly.

-In the Methods, please be more specific regarding the inclusion and exclusion criteria.

Response: Dear reviewer, thank you for your comment. The manuscript was changed accordingly.

-Please inspect the manuscript carefully for minor typos, language and grammatical errors.

Response: Dear reviewer, thank you for your comment. We revised the manuscript regarding the grammatical and language errors carefully.

-Please try to improve the quality of Tables and Figures in terms of font size and style.

Response: Dear reviewer, thank you for your comment. Due to the great extent of content in the tables, we had to use a smaller font. However, we tried to make the style of the tables and figures more appropriate.

Reviewer 3 Report

This systematic review described demographic epidemiologic and clinical features of MPX disease. Although it is quite complete, some issues are arising.

1) It is not well clarified the criteria selection of the studies, since the references cited in the text mainly referred to a period before May 2022.  Only 17 studies (including WHO website) published in 2022 are included in the references.

Therefore I suggest to revise the literature and update the references and the tables. 

2) Demographic and epidemiologic data are too concise, particularly if they referred to 2022. Please add more informations. 

3) As of May 2022, numerous cases of MPX occurred in non-endemic countries. It should be better explained whether clinical differences could be evidenced in the cases before and after 2022. 

4) Ocular manifestations have been frequently associated with severe outcomes. This should be added with references in the text.

5) There is little discussion that both genders can be infected with MPX.  This is an important concept for reducing stigma. Please cite recent cases of MPX in women, including old women. 

6) Please use MSM (men who have sex with men)  instead of gays

Author Response

Dear reviewer,

Thank you for your thoughtful comments. We have carefully edited the manuscript as you requested and provided a point-by-point response below.

1) It is not well clarified the criteria selection of the studies, since the references cited in the text mainly referred to a period before May 2022.  Only 17 studies (including WHO website) published in 2022 are included in the references. Therefore, I suggest to revise the literature and update the references and the tables. 

Response: Dear reviewer, we updated our review and extended it to the scientific articles that were published from May 1, 2022, to February 15, 2023. We included 44 more studies, 42 of which were according to the 2022 outbreak. Also, we performed a meta-analysis on the data obtained from both pre-and post-2022 outbreaks to highlight the differences in mpox characteristics over two periods. References and the tables were also updated accordingly.

2) Demographic and epidemiologic data are too concise, particularly if they referred to 2022. Please add more information.

Response: Dear reviewer, thank you for your comment. In the Results and Discussion sections, we added more information about the epidemiologic and demographic characteristics of mpox cases.

3) As of May 2022, numerous cases of MPX occurred in non-endemic countries. It should be better explained whether clinical differences could be evidenced in the cases before and after 2022. 

Response: Dear reviewer, we updated the manuscript and performed a meta-analysis on the data obtained from both pre-and post-2022 outbreaks to highlight the differences in mpox characteristics over two periods.

4) Ocular manifestations have been frequently associated with severe outcomes. This should be added with references in the text.

Response: Dear reviewer, thank you for your comment. We added an extra paragraph with references in the Discussion part. Also, we conducted a meta-analysis on the patients' ocular manifestations, which is demonstrated in the Results section.

5) There is little discussion that both genders can be infected with MPX.  This is an important concept for reducing stigma. Please cite recent cases of MPX in women, including old women. 

Response: Dear reviewer, thank you for your thoughtful comment. We added some more discussion on this issue. Also, related references were included in the Discussion section.

6) Please use MSM (men who have sex with men) instead of gays.

Response: Dear reviewer, thank you for pointing out this issue. The manuscript was changed accordingly.

Round 2

Reviewer 1 Report

the revised manuscript is now relevant with the aim of the study since  pubmed database now include 2023

Reviewer 3 Report

The authors have exhaustively answered the queries. The review is now complete and updated.